



# The Influence of Glacial Northern Hemisphere Ice Sheets on Atmospheric Circulation

Himadri Saini[1,2,3], David K. Hutchinson[1,2], Josephine R. Brown[3,4], Russell N. Drysdale[3], Yanxuan Du[3,4], and Laurie Menviel[1,2]

[1]Climate Change Research Centre, University of New South Wales, Sydney, New South Wales, Australia
[2]The Australian Centre for Excellence in Antarctic Science, University of Tasmania, Hobart, Tasmania 7001, Australia
[3]School of Geography, Earth and Atmospheric Sciences, University of Melbourne, Melbourne, Victoria, Australia
[4]Australian Research Council Centre of Excellence for Weather of the 21st Century, University of Melbourne, Melbourne, Victoria, Australia,

**Correspondence:** Himadri Saini (himadri.saini@unimelb.edu.au)

**Abstract.** During the last glacial period, Northern Hemisphere (NH) ice sheets significantly influenced atmospheric circulation, yet their broader impacts beyond the North Atlantic remain underexplored. Using the Australian Earth System Model (ACCESS-ESM1.5), we simulate a glacial climate around 49,000 years ago (49 ka) during Marine Isotope Stage 3 (MIS 3, 65-25 ka), a period marked by prominent millennial-scale variability. Our findings demonstrate that the NH ice sheets induced

a southward shift of the NH westerlies, increasing rainfall over Eurasia during boreal summer while reducing it in winter. The ice sheets' influence also extended across the tropics and the Southern Hemisphere (SH), driving a southward displacement of the Intertropical Convergence Zone (ITCZ) and the NH Hadley cell during austral summer, intensifying Australian rainfall. Additionally, including the Laurentide and Antarctic ice sheets (LIS and AIS) shifts the SH Hadley cell and the SH westerlies equatorward during JJA. These findings highlight the non-linear interactions between ice sheets, large-scale atmospheric

circulation, and precipitation patterns.

## 1 Introduction

Earth's climate history has been shaped by the expansion and retreat of Northern Hemisphere (NH) ice sheets, particularly during the last glacial cycle. The expansion of large NH ice sheets has long been associated with global and regional cooling, particularly over the North Atlantic, which is directly influenced by their extent (Kageyama et al., 2020). Ice sheet growth

affects land-ocean geography by lowering sea levels (Lambeck et al., 2014), exposing land bridges, and altering local heating patterns due to the differing heat capacities of land and ocean (Byrne and O'Gorman, 2013). Additionally, changes in land bridges and oceanic gateways, such as the Bering Strait, influence large-scale ocean circulation, including the Atlantic Meridional Overturning Circulation (AMOC) (Hu et al., 2015). These factors have significant implications for the climate system, particularly through their impact on atmospheric circulation and ocean currents.

In today's climate, warm water in the North Atlantic is transported northward via the Gulf Stream, and atmospheric Rossby wave propagation results in a northeastward-tilted Atlantic jet stream. This configuration contributes to the relatively mild win-





ters in Europe compared to regions at similar latitudes in North America. However, during past glacial periods, and particularly the Last Glacial Maximum (LGM; 19-21 ka), thick ice sheets covered northern North America and may have reached up to 5 km in height (Peltier, 2004). This so-called Laurentide Ice Sheet (LIS) caused a pronounced southward shift of the jet stream,

and in some cases, a more zonal orientation compared to today's tilted pattern (Pausata et al., 2011; Löfverström et al., 2014; Ullman et al., 2014). This southward shift, which was primarily driven by ice sheet topography, rather than changes in albedo or greenhouse gas (GHG) forcing (Pausata et al., 2011), had important consequences for European climate, impacting winter storm tracks, precipitation, and seasonal temperatures (Löfverström et al., 2014).

Previous research on understanding the impact of NH ice sheets on climate (Pausata et al., 2011; Löfverström et al., 2014;

Ullman et al., 2014; Wang et al., 2018; Izumi et al., 2023) focused on the LGM, an important period due to its extreme climate conditions, including the coldest global temperatures (Kageyama et al., 2020), lowest atmospheric $CO_2$ levels (Köhler et al., 2017), and most extensive ice sheets (Peltier, 2004; Clark et al., 2009; Kageyama et al., 2017; Gowan et al., 2021). Most of these studies examined the North Atlantic region, assessing the impact of ice sheets on westerlies and the Atlantic jet, with limited investigation into regions beyond this sector. An exception is the work of Löfverström et al. (2014), who investigated

the LGM as well as pre-LGM ice sheets of Marine Isotope Stage (MIS) 5b (∼88 ka) and MIS 4 (∼66 ka). Their study found that the westerlies shifted southward in all ice sheet configurations, but a fully zonal jet orientation emerged only during the LGM. They also suggested that the ice sheet topography dynamically induced warming in Alaska and central Asia, potentially contributing to the westward expansion of the Eurasian ice sheets from MIS 4 to the LGM. In addition, Ullman et al. (2014) and Zhang et al. (2014) examined the impact of changes in the height of the LIS on the atmospheric-oceanic system.

Despite these advances, the climatic impact of NH ice sheets during MIS 3 remains poorly constrained, likely due to the significant millennial-scale variability characterizing this period. Additionally, studies exploring their broader climate impacts outside the North Atlantic remain limited.

Here, we address these gaps by examining the atmospheric and climatic impacts of NH ice sheets during MIS 3. MIS 3 is characterized by pronounced millennial-scale climate variability, including abrupt temperature fluctuations in Greenland of

8–10°C (Huber et al., 2006). This period was also associated with multiple episodes of AMOC weakening (Menviel et al., 2020). Within MIS 3, the period around 49 ka stands out due to one of the highest obliquity values (24.43°) of the last glacial cycle (Berger, 1978), resulting in greater summer insolation in both hemispheres compared to the preindustrial (PI). In particular, 49 ka exhibits an obliquity value higher than some interglacial periods, such as the Last Interglacial (LIG, 127 ka), yet features a climate state marked by low $CO_2$ (∼199 ppm, similar to LGM (Köhler et al., 2017)) and extensive Laurentide

and Scandinavian ice sheets (Gowan et al., 2021), leading to a colder-than-PI climate. Furthermore, global sea levels were approximately 60-65 m lower than PI (Shakun et al., 2015). This combination of factors makes 49 ka an intriguing period for assessing how ice sheets influenced atmospheric circulation and hydroclimate on a planetary scale.

Here, we use the Australian Earth System Model (ACCESS-ESM1.5) to examine the influence of glacial ice sheets on global atmospheric circulation, focusing on their cascading effects across Eurasia, the tropics, and the Southern Hemisphere

(SH). Our analysis emphasizes the role of the ice sheets in modulating Rossby waves, Hadley circulation, and the Intertropical Convergence Zone (ITCZ), shaping rainfall patterns in both hemispheres, and influencing regional climates, such as over



Eurasia and northern Australia. In addition, we examine how the inclusion of the larger MIS 3 ice-sheets compared to PI influence SH westerlies. By exploring these interconnected processes, we provide insights into how large-scale climatic drivers may have contributed to shaping human migration and settlement patterns during this dynamic period.

## 2 Methods

### 2.1 Model description

We employ the Australian Community Climate and Earth System Simulator-Earth System Model version 1.5 (ACCESS-ESM1.5), a comprehensive Earth system model developed for CMIP6 (Ziehn et al., 2020). The atmospheric model is the UK Met Office Unified Model version 7.3 (UM7.3) (Martin et al., 2010; The HadGEM2 Development Team et al., 2011) with a horizontal resolution of $1.875° \times 1.25°$ and 38 vertical levels. The land surface is represented by the Community Atmosphere Biosphere Land Exchange (CABLE) model version 2.4 (Kowalczyk et al., 2013), which includes biogeochemical processes through the CASA-CNP module (Wang et al., 2010). The ocean component uses the NOAA/GFDL Modular Ocean Model version 5 (MOM5) (Griffies, 2012), featuring a horizontal resolution of $1° \times 1°$, with the meridional spacing refined to $0.4°$ in the Southern Ocean and $0.33°$ near the equator, with 50 vertical levels. Furthermore, sea ice processes are simulated using the Los Alamos National Laboratory Consortium Model for Sea Ice Development (LANL CICE) version 4.1 (Hunke et al., 2010). The coupling between the atmosphere, ocean and sea ice components is achieved through the Ocean Atmosphere Sea Ice Soil–Model Coupling Toolkit (OASIS-MCT) (Craig et al., 2017). The ocean carbon cycle dynamics are modeled using the nutrient-phytoplankton-zooplankton-detritus (NPZD) scheme in the World Ocean Model of Biogeochemistry and Trophic dynamics (WOMBAT) model for ocean biogeochemistry (Oke et al., 2013).

### 2.1.1 Experimental setup

The pre-industrial (PI) simulation is derived from a climate scenario based on the year 1850 CE (piControl, doi:10.22033/ESGF/CMIP6.4248) as described by Ziehn et al. (2020). Orbital parameters and GHG concentrations for the PI configuration are provided in Table 1. This simulation has been integrated for 1000 years. We use the average of the last 100 years of this simulation (Figure A1, cyan line).

To investigate the climate around 49 ka, the boundary conditions are changed in a step-wise manner using the ACCESS-ESM1.5 model. The first experiment, 49ka-co, simulates a baseline climate with orbital parameters (Berger, 1978) and GHG concentrations (Köhler et al., 2017) corresponding to 49 ka (Table 1). 49ka-co is run for 618 years (Figure A1, black line).

In a second step, 49ka-ice, we incorporate the ice sheet extent from Gowan et al. (2021) corresponding to 52.5 ka, adopting the maximal scenario with an ice-capped Hudson Bay (Figure 1). This time period was chosen because the simulated ice sheet at 52.5 ka most closely matches the sea-level estimates for 48–52 ka, based on global seawater $\delta^{18}O$ stack records (Shakun et al., 2015). The vegetation cover is modified to reflect ice coverage by removing vegetation in ice-covered regions and converting needle leaf evergreen forests in areas between the Cordilleran and LIS to bare soil (Figure 1). These modifications



|  | PI | 49ka-full |
|---|---|---|
| Orbital parameters | | |
| Eccentricity | 0.01674 | 0.01292 |
| Obliquity (°) | 23.459 | 24.435 |
| Perihelion-180 (°) | 100.33 | 62.451 |
| Greenhouse Gases | | |
| $CO_2$ | 284.3 | 199 |
| $N_2O$ | 273 | 237 |
| $CH_4$ | 284.3 | 432 |

**Table 1.** Orbital parameters and Greenhouse gas concentrations for PI and 49 ka climate.

| Experiment | Model years | Years analysed | Orbital parameters and GHGs | Albedo and vegetation | Salinity | Topography and runoff |
|---|---|---|---|---|---|---|
| 49ka-co | 0 to 618 | 568 to 618 | 49ka | PI | PI | PI |
| 49ka-ice | 297 to 824 | 519 to 569 | 49ka | 49ka | PI | PI |
| 49ka-full | 824 to 1555 | 1451 to 1551 | 49ka | 49ka | 49ka | 49ka |

**Table 2.** Boundary conditions used in all 49 ka experiments. Model years indicate the total number of years that the corresponding experiment has run for in the full time series shown in Figure A1.

align with the vegetation reconstruction from Allen et al. (2020) for 52 ka. Additionally, C3 crops are removed, and replaced by the next two most dominant vegetation types in the corresponding grid cell. For example, in India, C3 crop values are replaced
90 by broadleaf deciduous and C4 grass. In Europe and eastern North America, they are replaced by broadleaf deciduous and C3 grass, while they are replaced by broadleaf deciduous over China. 49ka-ice (Figure A1, orange line) starts from yr 297 of 49ka-co. As can be seen in Figure A1, 49ka-ice is not in complete equilibrium.

In the 49ka-ice experiment, additional modifications are introduced at different stages (Figure A1, orange line). At year 824, the topography is adjusted to reflect conditions at 49 ka, based on Gowan et al. (2021), including an increase in ice sheet height
95 and changes to the land-sea mask, such as the closure of the Bering Strait (done in the previous step), Hudson Bay, the Sahul and Sunda shelves, and the Tasman Gateway (Figure 1). Nearest-neighbour interpolation is applied to determine vegetation types on newly exposed land. The final experiment, incorporating all these changes, is referred to as 49ka-full (Figure A1, blue line, Table 2) and was run for 292 years. In the Results section, we compare the average of the last 100 years of 49ka-co, last 50 years of 49ka-ice and the average of the last 100 years of the 49ka-full experiment (Table 2, Figure A1, darker colors).
100 The ITCZ position is determined using the precipitation centroid index (Braconnot et al., 2007), which identifies the latitude







**Figure 1.** Vegetation and ice mask changes at 49 ka compared to PI. The color indicates the dominant Plant Functional Type (PFT) per grid cell.



that evenly divides tropical precipitation between 20°S and 20°N. This approach effectively represents the center of mass of tropical rainfall distribution. The strength of the Northern Hadley cell is determined by the maximum value of the atmospheric mass streamfunction within the 400–600 hPa pressure range, while the strength of the Southern Hadley cell is determined by its minimum value within the same range.

## 3 Results

### 3.1 Simulated glacial climate at 49 ka compared to PI

We first describe the 49ka-full experiment in comparison to PI. At 49 ka, the higher obliquity (24.43°) compared to PI (23.45°) results in slightly increased summer insolation in both hemispheres (Figure A2). Despite the higher obliquity, the glacial simulation (49ka-full) exhibits substantial oceanic and atmospheric cooling due to lower GHG concentrations and extensive ice sheet expansion at this time (Figure 2). Global annual mean surface air temperatures (SATs) at 49 ka are 2.7°C lower than PI, with the NH cooling by 4.9°C and the SH by 2°C (Figure 2a). The most pronounced cooling is simulated in ice sheet regions and northern high latitudes. Seasonal variations are evident in the NH, where temperatures drop by 6.4°C in winter (DJF) and 2.6°C in summer (JJA). Over SH, DJF (austral summer) SATs are 1.7°C lower, while JJA (austral winter) SATs are 2.4°C lower.

Global mean ocean temperature is 1°C lower, and global annual mean sea surface temperature (SST) (Figure 2b) is reduced by 2°C. Higher SSTs in the Labrador region are linked to a stronger AMOC in 49ka-full, with a transport rate of 31 Sv (about 10 Sv greater than the PI value). This strengthening is attributed to topographical changes and the presence of ice sheets, which enhance westerly winds over Baffin Bay and weaken them over the Labrador Sea (Figure A3). These wind changes modify the North Atlantic gyres, leading to an overall strengthening while shifting the separation between the subpolar and polar gyres southward and orienting the subpolar gyre in a more zonal direction (Figure A3). A strengthening of the cyclonic circulation in the Labrador Sea is also simulated, which increases the westward transport of warm water from south of Iceland to the Labrador Sea and further strengthens the AMOC. The stronger AMOC also influences sea ice extent. During boreal winter, the sea-ice edge at 49 ka expands in the North Pacific and Nordic Sea sectors but retreats in the Labrador Sea due to the influx of warm water (Figure 2b, orange dashed line compared to black dashed line). In boreal summer, the 49 ka sea-ice edge is similar to PI across most regions, except in the Nordic seas, where the sea-ice edge extends further south than in PI by 5 to 6° (Figure 2b, orange solid line compared to black solid line). In the SH, austral summer sea ice retreats in the South Atlantic as stronger North Atlantic Deep Water (NADW) leads to a deep-ocean warming of about 0.3°C (not shown), and this warm water gets upwelled in the South Atlantic (Figure 2b, orange dashed line compared to black dashed line). However, the austral winter sea ice edge remains similar to PI in the South Atlantic while expanding in the Indian and Pacific sectors of the Southern Ocean (Figure 2b, orange solid line compared to black solid line).

Drier conditions are simulated at 49 ka compared to PI over NH high-latitude land regions, particularly over the LIS, due to colder conditions and reduced atmospheric moisture availability (Figure 2c). However, precipitation increases in the North



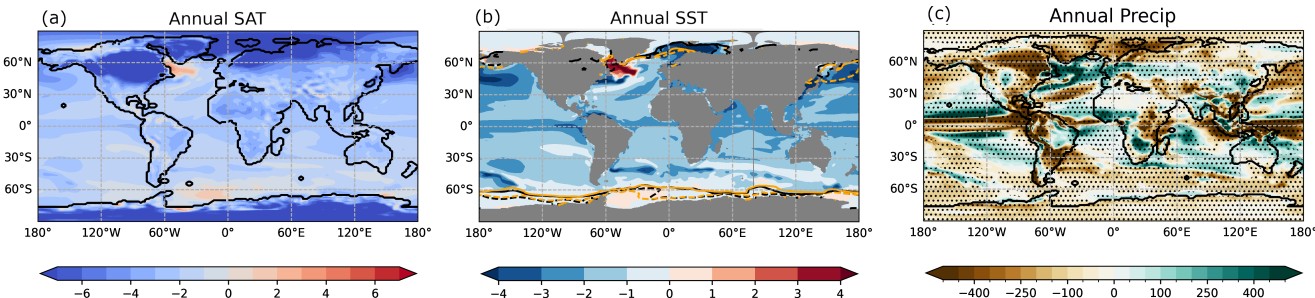

**Figure 2.** Annual mean surface air temperature (SAT), sea surface temperature (SST), and precipitation anomalies for 49 ka climate (exp 49ka-full) compared to PI. Contours are seasonal mean 15% ocean ice concentration for (orange) 49 ka and (black) PI during DJF (dashed) and JJA (solid). Stippling in subpanel (c) indicates significant changes based on a Student's t-test at 95% significance level.

Atlantic and Eurasia, between 30°N and 60°N, as well as over India and East Asia. In contrast, most land areas in the SH display decreased precipitation, except over northern Australia, parts of Southeast Africa, and over eastern Brazil.

## 3.2 Impact of the ice sheet topography on the North Atlantic and Eurasia

At 49 ka, the presence of large ice sheets over North America significantly alters the atmospheric circulation, particularly in the North Atlantic region. In DJF, the westerlies over the North Atlantic strengthen due to the strong temperature gradient and shift ∼6° southward compared to PI (Figures 3a,c and 4a), with a more zonal flow replacing the northeast-southwest tilt of the jet stream (Figure A4). Geopotential height (GH) differences (Figure 3e) indicate that the LIS induces a high-pressure system over North America, initiating a Rossby wave response. This Rossby wave causes a southward shift of the Icelandic low and a weakening of the Azores High, contributing to the zonal orientation of the westerlies (Figures 3c, A4, and A5). The southward shift and zonal jet alter moisture-bearing storm tracks, which in PI were tilted northeast-southwest, to a more zonal configuration at 49 ka. This reorganization of pressure systems, with a negative pressure anomaly extending from the North Atlantic to eastern Eurasia (Figure 3e), alters precipitation patterns, enhancing precipitation over the North Atlantic and western Europe (Figure 3g). However, colder conditions over Eurasian landmasses lead to drier conditions inland in DJF, despite the persistence of negative pressure anomalies (Figure 3a,e,g).

The westerlies are also shifted southward in JJA, with the flow remaining more zonal than during PI (Figures 3d, A4, and 4b), coinciding with a positive pressure anomaly over the Labrador Sea region while a negative pressure anomaly forms over Europe (Figure 3f). These pressure anomalies enhance the westerly flow and contribute to moisture convergence toward the Eurasian landmass, increasing precipitation along the westerly band in the North Atlantic Ocean and over parts of Eurasia (Figure 3d,f,h). In addition, mild temperature conditions over Eurasia (Figure 3b) further contribute to the increased precipitation. Precipitation also increases north of 60°N in Asia (Figure 3h) due to higher specific humidity resulting from warmer conditions and the development of a low-pressure system (Figure 3b,f). The warming in this region arises from the topographical changes corresponding to 49 ka, where ocean points have been converted to land. Since land has a lower heat capacity than the ocean,



it warms more rapidly in summer, amplifying the temperature-driven moisture increase. Conversely, north of 60°N in Europe precipitation decreases. This is due to the influx of colder north-easterly winds and the presence of a blocking high over the Labrador-Greenland sector, which suppresses rainfall in these areas.

In the 49ka-co experiment, strong cooling reduces moisture availability, leading to widespread drying over NH landmasses during DJF (Figure 5a,c). The westerlies are slightly enhanced and shift southward by only ∼1° (Figure 4a). This shift is accompanied by a low-pressure anomaly over the North Atlantic and a high-pressure system over Europe, which enhances subsidence and suppresses precipitation over Europe (Figure 5b,c). This pattern is consistent with the overall drying trend in the NH, except for a small increase in precipitation over the North Atlantic Ocean (Figure 5c), likely due to some storm activity in this region.

Changing the vegetation and albedo in 49ka-ice has little impact compared to 49ka-co during DJF, except over northern North America (Figure 5d-f), where drying is slightly enhanced due to the albedo change over the LIS. The westerlies intensify further compared to 49ka-co, but their position shifts back to the PI configuration (Figure 4a). The GH response during DJF features a low-pressure anomaly over the Hudson Bay and Labrador Sea, which is consistent with increased precipitation in this region (Figure 5e,f).

In JJA, the 49ka-co experiment shows no change in westerly strength and position over the North Atlantic (Figure 6b, 4b) and modest temperature and precipitation changes over the NH landmass (Figure 6a,c). In contrast, the 49ka-ice experiment exhibits pronounced drying over the Laurentide region and northern Europe, driven by intense cooling from albedo changes and persistent high pressure over these regions (Figure 6d-f). While the westerlies strengthen in 49ka-ice (Figure 4b), their position remains unchanged. This intensification leads to slightly wetter conditions inland over Eurasia in the 49ka-ice scenario.

Figures 5, 6, and 4a show that only the full implementation of the ice sheet, including its topography and height, can induce a significant planetary wave response in the atmosphere, with southward-shifted North Atlantic westerlies and larger temperature and precipitation anomalies. This response results in substantial drying over the northern high-latitude landmasses, including the Laurentide region during DJF, along with increased milder and wetter conditions over Eurasia during JJA.

### 3.3 Impact of the ice sheet topography on Hadley cell, ITCZ, and SH westerlies

The presence of the LIS forces a southward shift of the Ferrell cell in DJF, driven by altered thermal and pressure gradients. This displacement further leads to a southward migration of the NH Hadley cell's southern edge, shifting from 7.5°S in PI to 11°S in 49ka-full (Figure 7a, see zero contour), and results in a globally averaged ITCZ shift of approximately 1.5° southward in DJF (Figure 3g). Additionally, the NH Hadley cell strengthens by 20 Sv relative to PI. The southward shift of the NH Hadley cell and ITCZ during DJF is also captured in 49ka-co (Figure A6a), where the NH Hadley cell strengthens by +11 Sv relative to PI. This shift is driven by peak summer insolation in the SH due to higher obliquity at 49 ka. However, the magnitude of change in 49ka-co is smaller than in 49ka-full (Figure 7a), with an ITCZ shift of 0.6°. Implementing albedo changes in 49ka-ice enhances the Laurentide cooling, further strengthening the NH Hadley cell (by +8 Sv) albeit without additional displacement (9.5°S in 49ka-co vs. 9.6°S in 49ka-ice) (Figure A6a,c). The minor shift in the NH Hadley cell and ITCZ position explains why lower latitude precipitation changes remain insignificant between 49ka-ice and 49ka-co (Figure 5c,f). However, the inclusion





**Figure 3.** (a,b) SAT (°C), (c,d) zonal winds at 850 hPa (m/s) overlaid with wind vectors, (e,f) geopotential height at 850 hPa (m) with zonal mean removed, and (g,h) precipitation (mm/yr) anomalies between 49ka-full and PI during (left) DJF and (right) JJA. Stippling in subpanels (g) and (h) indicates significant changes based on a Student's t-test at 95% significance level.





**Figure 4.** Zonal winds (m/s) averaged (top) over the North Atlantic (80W:0E) and (bottom) averaged over all longitudes representing westerlies over the SH during (a,c) DJF and (b,d) JJA seasons.

of topography in 49ka-full results in a slight additional strengthening of the NH Hadley cell and a more substantial southward
ITCZ shift (by 1.5° relative to 49ka-ice), due to the stronger temperature gradient between the two hemispheres.

During JJA, the southern edge of the SH Hadley cell shifts northward by 2° in 49ka-full, moving from 32°S in PI to 30°S,
along with a northward shift of the Southern Ferrel cell (Figure 7b, see zero contour). This shift causes a slight northward
migration of the globally averaged ITCZ (Figure 3h), while the SH Hadley cell strengthens by 35 Sv. In 49ka-co, a slight
warming over the North Atlantic and stronger cooling in the SH shifts the SH Hadley cell's southern edge to 31°S, accompanied
by a strengthening of +22 Sv (Figure A6b). The changes in 49ka-ice do not alter the SH Hadley cell's latitudinal position but
weaken its strength to PI levels (Figure A6d), due to intensified Laurentide cooling. As a result, precipitation changes in the
lower latitudes remain small (Figure 6c,f). However, incorporating topography introduces localized warming in Siberia and



## 49ka-co - PI

## 49ka-ice - 49ka-co

## 49ka-full - 49ka-ice

**Figure 5.** DJF (a,d,g) SAT (°C), (b,e,h) geopotential height (m) overlaid with winds at 850 hPa and precipitation (mm/yr) anomalies between (a-c) 49ka-co minus PI, (d-f) 49ka-ice minus 49ka-co, (g-i) and 49ka-full minus 49ka-ice. Stippling indicates significant changes based on a Student's t-test at 95% significance level.

other regions where ocean grid cells are replaced by land, while the addition of AIS topography induces a strong Antarctic cooling, amplifying hemispheric temperature contrasts. This enhanced thermal gradient leads to a slight northward ITCZ shift,
further strengthening the SH Hadley cell and shifting it an additional 1° northward relative to 49ka-ice.

The SH westerlies also exhibit distinct responses across the simulations. In DJF, they weaken slightly in 49ka-co but weaken further in 49ka-ice, accompanied by a small northward shift (Figure 4c). In contrast, in 49ka-full, they remain in the same position as in PI but strengthen due to the increased height and extent of the AIS. During JJA, the SH westerlies weaken across all 49 ka simulations compared to PI, with the strongest weakening occurring in the experiment where albedo is changed
(Figure 4d). However, in 49ka-full, they shift northward by 2.5°, a displacement linked to the northward shift of the SH Hadley cell in 49ka-full.



## 49ka-co - PI



## 49ka-ice - 49ka-co

## 49ka-full - 49ka-ice

**Figure 6.** Same as Figure.5 but for JJA.

### 3.4 Impact on Australian rainfall

As shown in Figure 3, the implementation of glacial ice sheets leads to a notable increase in Australian rainfall during the monsoon season. This section focuses on the large-scale processes driving this regional rainfall response. While a detailed analysis of monsoon changes in other regions lies beyond the scope of this study, our results underscore the broader influence of Northern Hemisphere ice sheets on hemispheric-scale atmospheric dynamics.

The presence of NH and larger Antarctic ice sheets induces substantial atmospheric circulation changes by modifying the Hadley and Ferrel cells and influencing the westerlies in both hemispheres. These shifts propagate through the atmosphere, altering surface pressure systems and triggering a hemispheric-scale response.

During DJF, the NH ice sheet forces a southward shift of the Hadley and Ferrel cells (Figure 7a), enhancing subsidence over East Asia around 40°N and leading to drier conditions (Figure 3e,g). This southward displacement of the NH Hadley cell extends across hemispheres, impacting Australian rainfall by shifting convergence southwards along with the ITCZ (Figure 7a).





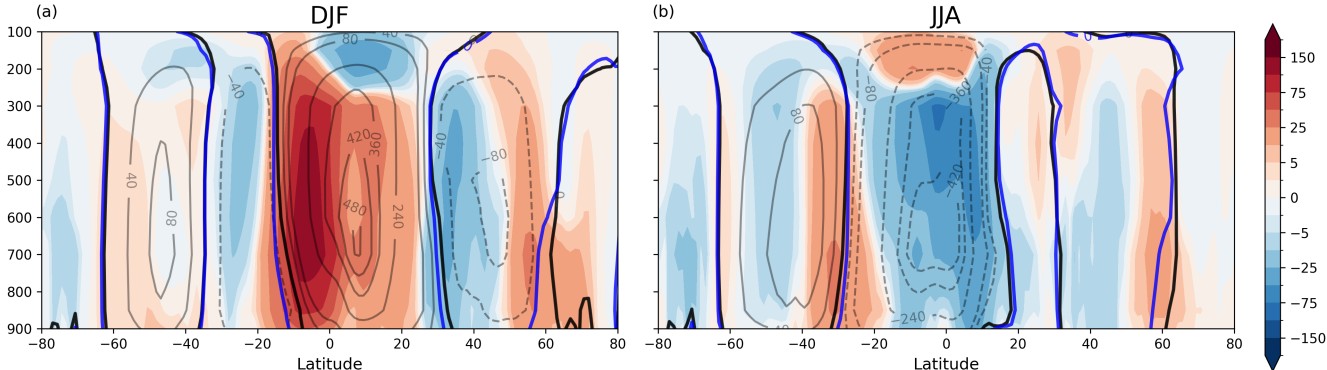

**Figure 7.** Atmospheric mass streamfunction (Sv) anomalies (shading) between 49ka-full and PI for (a) DJF and (b) JJA. Contours (solid=positive; dashed=negative) are PI (black) absolute streamfunction values. The thick black and blue lines represent the zero contour for PI and 49ka-full, respectively.

These changes explain the wetter conditions in northern Australia and drier conditions over the Maritime Continent (Figure 3g). A comparable but weaker response is simulated in the 49ka-co experiment (Figure 5c). In contrast, changes in albedo do
not impact Australian precipitation significantly (Figure 5f). However, modifying the topography induces wetter conditions in northern Australia driven by shifts in the Hadley cell and ITCZ, while southern Australia experiences drier conditions (Figure 5h,i).

During JJA, most of Australia experiences drying (Figure 3h), primarily due to the establishment of blocking high-pressure systems south of the continent (Figure 3f). These high-pressure systems reinforce subsidence and restrict moisture transport
into the region. However, the drying response is not significant over central Australia and remains less pronounced along the eastern and western edges of the continent. The southwestern part of Australia becomes wetter, while the southeastern region experiences drying, due to the northward displacement of the westerlies, resembling a negative winter Southern Annular Mode (SAM). Significant drying also occurs across northern Australia and much of the Maritime Continent south of the equator.

When only orbital parameters and GHGs are modified, the SH Hadley cell shifts northward and strengthens. However, in
this scenario, the weakening of the SH westerlies and the southward displacement of subtropical high-pressure systems result in increased rainfall over southern Australia (Figure 6b,c). Introducing albedo changes alone has little impact on Australian climate (Figure 6d-f). However, when ice sheet topography is incorporated, the northward shift in SH westerlies linked with the strengthening and equatorward displacement of the SH Hadley cell causes enhanced subsidence and thus drier conditions over Australia, coinciding with the northward ITCZ migration (Figure 6i, 4d). Compared to the PI climate, this drying effect is
somewhat mitigated, as the westerlies — despite their northward shift— remain weaker overall.





## 4 Discussion

Our simulations of the 49 ka glacial climate reveal a complex interplay of ice sheet topography, GHG forcing, and orbital configurations. The results underscore how large-scale climatic features, such as atmospheric circulation patterns, precipitation distribution, and ocean circulation, respond to glacial boundary conditions.

The global cooling in 49ka-full relative to PI (a 2°C reduction in global SATs) aligns with the expected glacial forcing associated with lower GHG concentrations and extensive ice sheet expansion. The pronounced winter cooling in the NH, exceeding 6°C, is consistent with PMIP4 simulations of LGM climate, with ice sheet-driven albedo changes amplifying high-latitude cooling during glacial periods (Kageyama et al., 2020).

**AMOC changes and Ice sheet influence**

Proxy records suggest significant variability in the AMOC during the MIS 3 period, with stronger AMOC during the onset of interstadials (Böhm et al., 2015; Henry et al., 2016; Menviel et al., 2020), and weak AMOC during stadials. The processes driving this AMOC variability is still debated (Menviel et al., 2020). The simulated AMOC in our model is strong at 49 ka (31 Sv). This strengthening is driven by changes in the LIS, which enhances westerly winds over Baffin Bay while weakening them over the Labrador Sea, intensifying the cyclonic circulation in the Labrador Sea and facilitating warm water transport to

the region. This process leads to sea ice retreat and increased salinity in polar waters, further reinforcing the AMOC. These findings align with earlier modelling studies emphasizing the role of topography and sea ice dynamics in AMOC enhancement (Pausata et al., 2011; Zhang et al., 2014; Hu et al., 2015; Sherriff-Tadano et al., 2018). For example, Pausata et al. (2011) demonstrated that the presence of an extensive ice sheet during the LGM enhanced AMOC by altering atmospheric circulation patterns. However, their simulations also showed that the cooling effects from lower GHG concentrations and increased albedo

suppressed AMOC strength. In our 49ka-full experiment, these cooling effects appear to be offset by high insolation, allowing topographical impacts to dominate.

Evidence based on foraminifer assemblages and mineral grains in ice-rafted debris also suggests a northward retreat of sea ice and higher temperatures in the Labrador Sea between ∼56 and 49 ka, potentially driven by the insolation maxima (Griem et al., 2019). After 49 ka, sea ice variability increased, with near-perennial sea ice cover during most stadials and seasonal ice-

free conditions during interstadials. However, it remains uncertain whether interstadials reflect a mean state or are primarily linked to DO variability.

**Ice sheet topography and NH westerlies**

The simulated changes in the North Atlantic westerlies and associated precipitation patterns during DJF highlight the critical role of ice sheet topography in driving atmospheric circulation (Figure 8a). At 49ka, the surface westerlies shift southward and

adopt a more zonal flow (Figure 3), aligning with previous studies (Löfverström et al., 2014; Ullman et al., 2014; Pausata et al., 2011; Kageyama et al., 2020; Stadelmaier et al., 2024), which emphasize the influence of glacial ice sheet topography on jet stream reorganization. Our results further expand on this, demonstrating that the high-pressure system induced by the LIS initiates a Rossby wave response, shifting the Icelandic low southward and weakening the Azores High. These circulation changes lead to reduced precipitation over NH landmass during DJF.



During JJA, the changes in westerlies and pressure result in increased rainfall over Eurasia (Figure 8b), with the jet stream extending further inland. Additionally, this rainfall increase is supported by milder land temperatures and enhanced moisture availability. Our findings on seasonal precipitation changes are consistent with those of Löfverström et al. (2014), as the ice sheet configuration used in our study closely resembles their MIS 4 ice sheet setup. However, while their study reports a full zonal orientation of the jet stream only during the LGM, our results demonstrate that this change can also occur with a MIS 3
ice sheet configuration, suggesting a more dynamic response to glacial ice sheets earlier in the glacial period.

### Meridional and Hemispheric Impacts of Ice Sheets

Previous studies have primarily focused on the impacts of NH ice sheets on the North Atlantic jet, with only one study (Löfverström et al., 2014) exploring precipitation changes over Eurasia. However, they used a slab ocean model and thus missed the atmosphere-ocean coupling feedback, which our study accounts for. Our results further demonstrate that the meridional
influence of glacial ice sheets extends far beyond the NH, impacting tropical climates, SH precipitation and surface westerlies.

The presence of ice sheet topography leads to a more pronounced southward shift of the NH Hadley cell during DJF and a stronger northward shift of the SH Hadley cell during JJA, compared to simulations that include only GHG concentrations, orbital parameters, and an ice mask at 49 ka (Figure 8a,b). This enhanced Hadley cell shift in 49ka-full drives a more substantial southward displacement of the ITCZ during DJF (Figure 8a). Consequently, the northern tropics experience drier conditions,
while precipitation increases in the southern tropics, consistent with the mechanisms proposed by Chiang and Bitz (2005).

During JJA, LIS topography induces warming over Siberia, in agreement with previous studies (Liakka and Lofverstrom, 2018; Bakker et al., 2020), which attribute this warming to large-scale atmospheric circulation changes driven by pressure differences linked to LIS height. In contrast, AIS topography enhances cooling over Antarctica during austral winter. This interhemispheric contrast, i.e. relatively warmer conditions in the NH and colder conditions in the SH, pushes the ITCZ north-
ward and strengthens the SH Hadley cell. Additionally, the SH Hadley cell northward shift is associated with the northward displacement of the SH westerlies (Figure 8b), consistent with Ceppi et al. (2013), who demonstrated that interhemispheric teleconnections link ITCZ migration with changes in Hadley cell strength and SH jet positioning. Likewise, NH cooling during DJF drives a southward shift of the North Atlantic westerlies, influencing the Hadley cell strength and position, and pushing the ITCZ closer to the equator.

These circulation shifts generate regional impacts. Increased subsidence over East Asia creates a high-pressure anomaly, leading to drier conditions, while the southward-shifted ITCZ enhances convergence over Australia, increasing rainfall. The larger shift in the ITCZ and the NH Hadley cell coincides with the maximum increase in Australian rainfall (Figure 8a).

The weakening of the SH westerlies during both DJF and JJA under a high-obliquity climate (49ka-co and 49ka-ice) compared to a low-obliquity climate (PI) aligns with findings from Timmermann et al. (2014). However, the increased height
and extent of the AIS counteract this weakening during DJF, leading to SH westerly intensification. Additionally, our results indicate that NH westerlies strengthen with albedo changes relative to orbital parameter and GHG adjustments, whereas SH westerlies weaken under the same conditions (Figure 4). Introducing topographical changes induces an equatorward shift in both NH and SH westerlies during their respective winter seasons (Figure 4a,d).

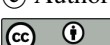



**Figure 8.** Summary of key atmospheric responses across experiments. (Top) DJF anomalies: (left) NH SAT (°C); (middle) latitudinal shifts (°) in westerlies averaged over the North Atlantic, NH Hadley cell's southern edge, and globally averaged ITCZ position; and (right) northern Australian precipitation (116°E–152°E, 25°S–12°S, mm/day). (Bottom) JJA anomalies: (left) NH SAT (°C); (middle) latitudinal shifts (°) in westerlies averaged over the North Atlantic, SH Hadley cell's southern edge (°), westerlies averaged over the SH; and (right) precipitation changes over Eurasia (60°W:80°E,48°N:60°N). Results are shown for the 49ka-co (cyan), 49ka-ice (orange), and 49ka-full (blue) experiments, with all values presented as anomalies relative to PI. Negative values in latitudinal shifts indicate an equatorward shift.



Overall, our study reveals nonlinear interactions between large-scale atmospheric circulation, surface westerlies, and precipi-
tation patterns under different climate boundary conditions (Figure 8). Orbital forcing, GHGs, albedo, and ice sheet topography
collectively shape these interactions in a nonlinear manner.

**MIS 3 climate in Australia**

Proxy records from marine sediments, lake sediments, and river systems, show that Australia experienced spatially variable
climates during MIS 3, with greater-than-modern mean water availability peaking between ∼49 and ∼40 ka (Kemp et al.,
2019). Wet conditions, especially in central and northern Australia between ∼50 and ∼45 ka, are linked to an intensified
Australian monsoon driven by the summer insolation peak. Our model results align with these observations, showing increased
annual precipitation over much of Australia at 49 ka, with particularly intensified monsoonal rainfall in the northern regions
(Figure 2c). This increase in rainfall is consistent with a southward displacement of the ITCZ due to the intensified NH Hadley
cell, driven by increased SH insolation during high obliquity. However, our results also reveal that the southward shift of
the ITCZ and Hadley cell occur in the 49ka-full simulation (which includes ice sheet topography) as well as in the 49ka-co
and 49ka-ice simulations (which lack ice sheet topography). Despite this, the intensified Australian monsoon rainfall is more
pronounced in the 49ka-full experiment, as the presence of ice sheets induces larger shifts of the Hadley cell and ITCZ.

Our simulations generally show an insignificant response in southern Australia at 49 ka, with some drying, which contrasts
with the increased rainfall seen in southern Australia in the Kemp et al. (2019) records. In their study, the strengthening
or northward movement of the southwesterly winds is linked to increased rainfall, a response that does not align with our
simulation. However, our results are somewhat consistent with Weij et al. (2024), in which speleothem growth rates from
southwest and southeast Australia indicate drier-than-modern conditions around 49 ka.

# 5 Conclusions

Our results underscore the far-reaching effects of NH ice sheets on both hemispheres, highlighting their critical role in shaping
regional climate patterns. We demonstrate that the presence of NH ice sheets strengthens the North Atlantic westerlies and
shifts them equatorward by 6° during DJF, leading to increased precipitation over the North Atlantic Ocean and enhanced
rainfall over the Eurasian landmass in boreal summer. These ice sheet-induced changes extend meridionally, with a southward
shift of the NH Hadley cell and ITCZ during DJF, resulting in increased Australian rainfall. Additionally, the inclusion of LIS
and AIS topography drives an equatorward shift of the SH westerlies and SH Hadley cell, further illustrating the interconnected
nature of atmospheric circulation changes driven by ice sheet topography.

*Data availability.* All final data from the modelling simulations will be published on the DRYAD repository under the corresponding
author's account (https://datadryad.org/search?utf8=%E2%9C%93&q=himadri+saini) and made publicly available upon acceptance of the
manuscript.



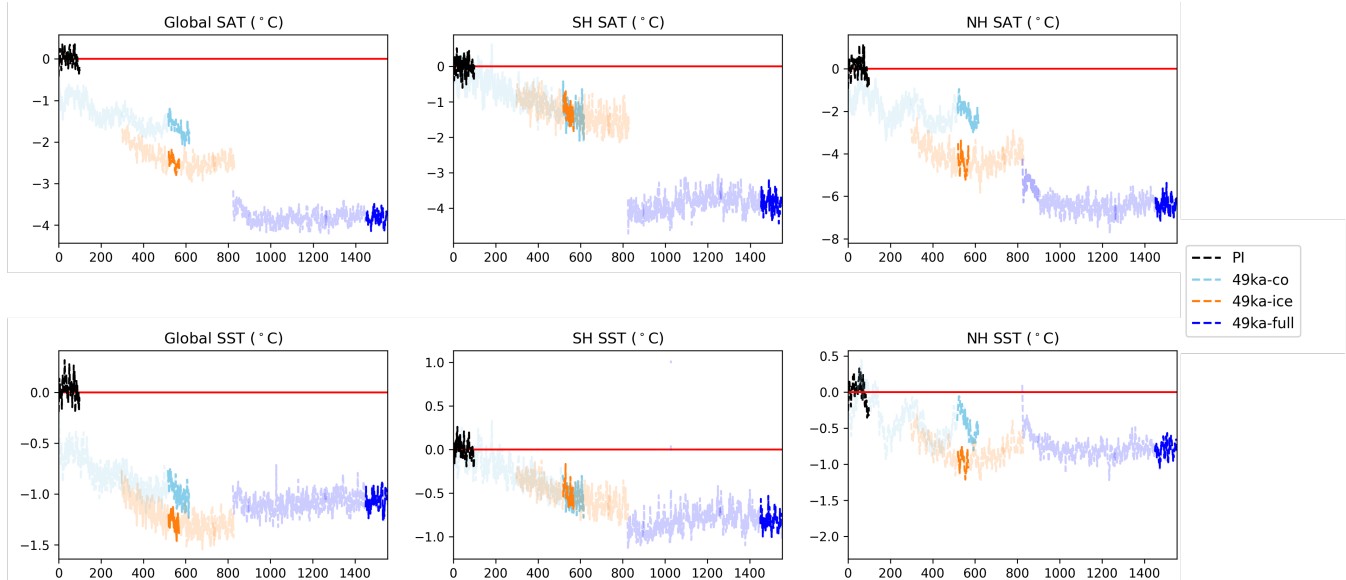

**Figure A1.** Time series anomalies of surface air temperature (SAT) for PI (black), 49ka-co (sky-blue), 49ka-ice (orange), and 49ka-full (blue) compared to PI. 49ka-co is orbital parameters+GHGs, 49ka-ice is 49ka-co+ albedo and vegetation changes. At year 547, surface salinity is gradually increased to account for the 44 m sea level drop relative to PI, based on the simulated ice sheet, achieving a global average increase of +0.33 psu from PI levels. At year 729, the Bering Strait is closed. At year 824, the 49 ka topography is implemented. Finally, at year 1258, river runoff is adjusted to reflect the 49 ka topography. Darker colors indicate the time-averaged period for each experiment used in the analysis. The horizontal red line is the zero line.

**Appendix A**




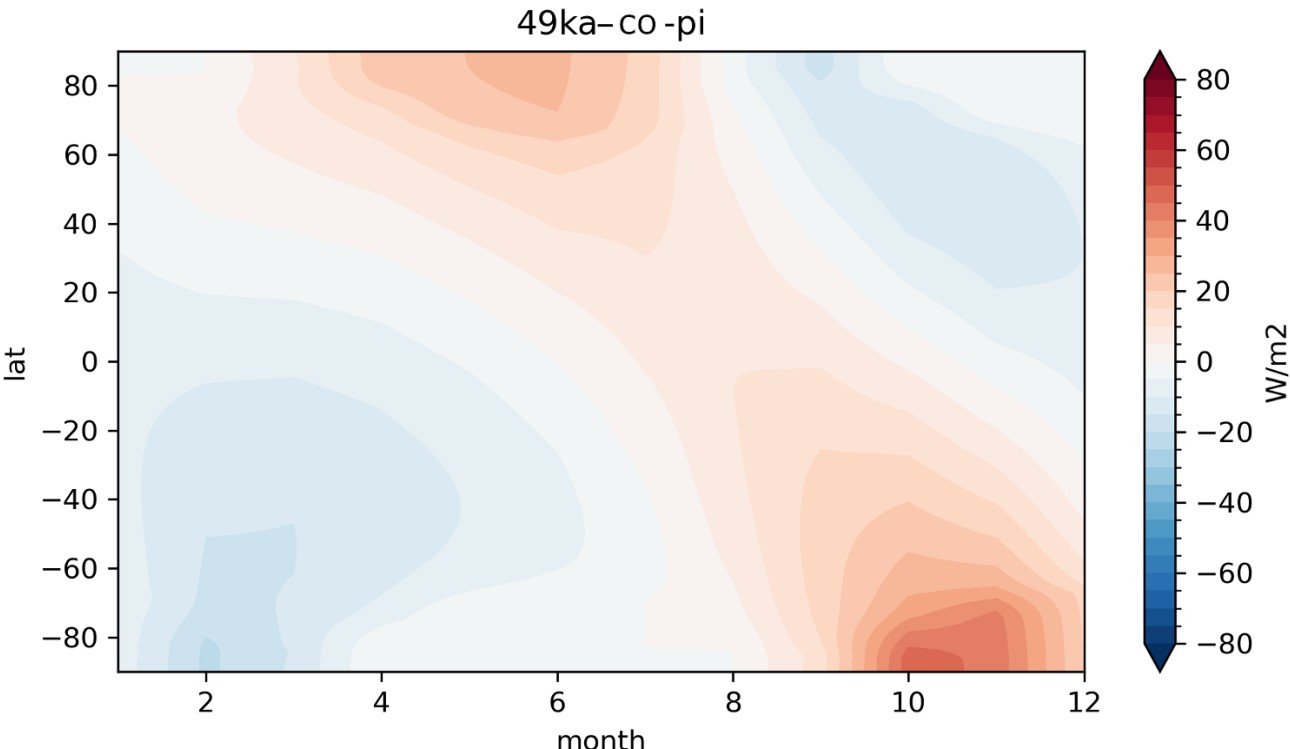

**Figure A2.** Insolation anomalies (W/m$^2$) between 49 ka and PI as a function of latitude and month.



**Figure A3.** (top) Winter mixed layer depth (m), (centre) Zonal winds (m/s) at 850 hPa, and (bottom) North Atlantic gyre strength (N/m2) for (left) PI and (right) 49 ka compared to PI.



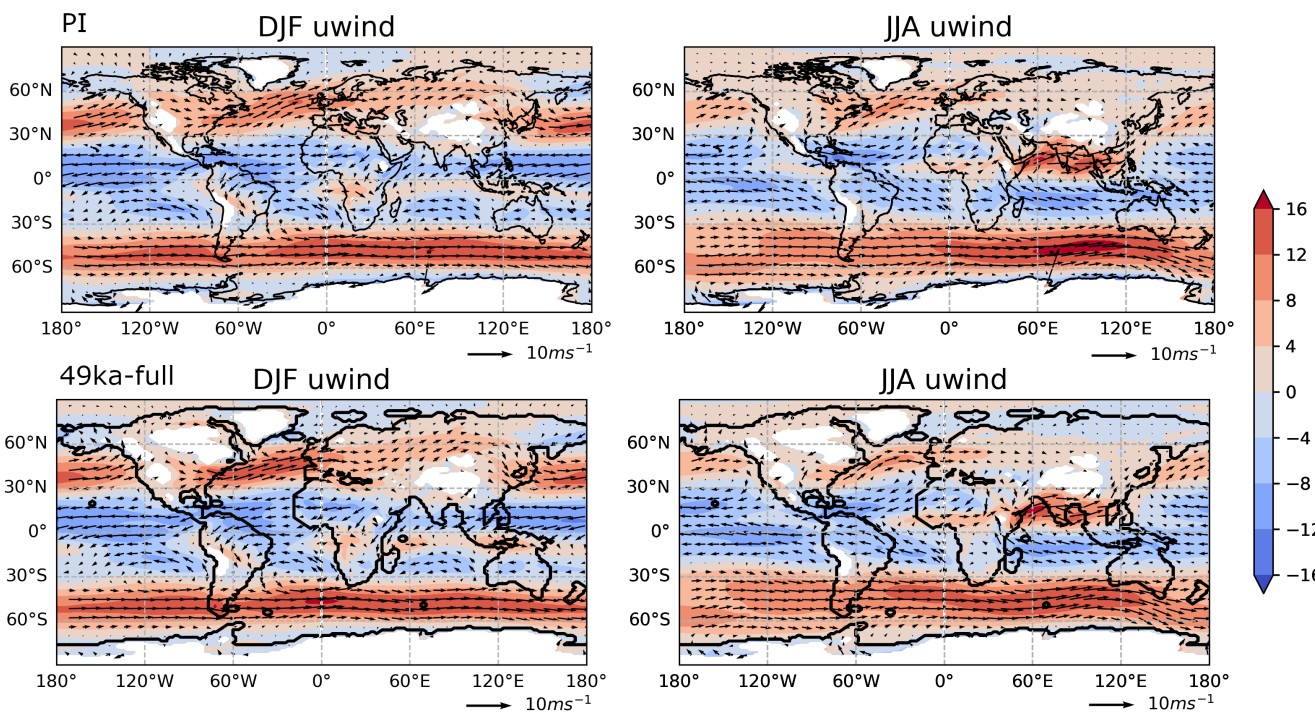

**Figure A4.** Zonal winds at 850 hPa (m/s) in (left) DJF and (right) JJA for (top) PI and (bottom) 49ka-full.

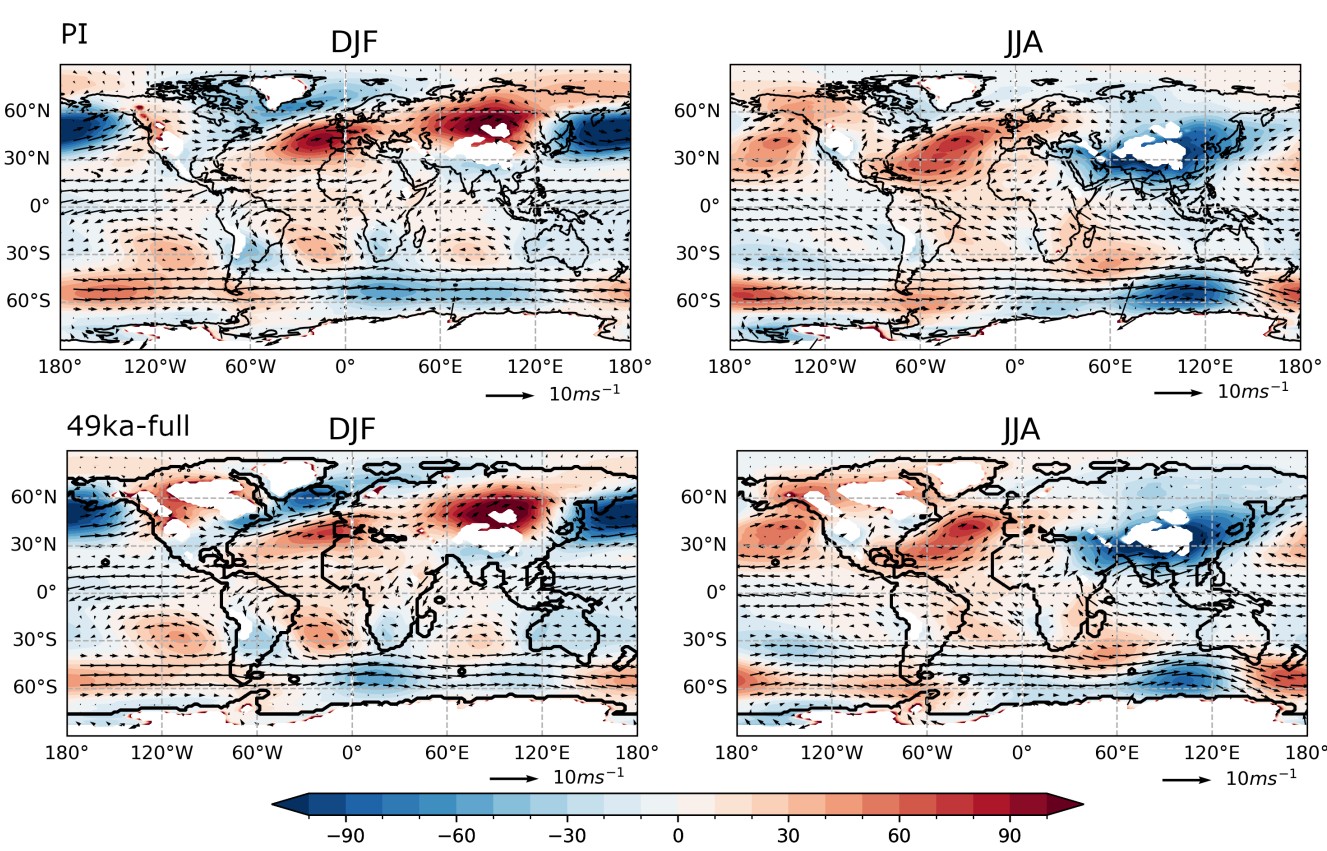

**Figure A5.** Geopotential height (m) at 850 hPa with zonal mean removed in (left) DJF and (right) JJA for (top) PI and (bottom) 49ka-full.



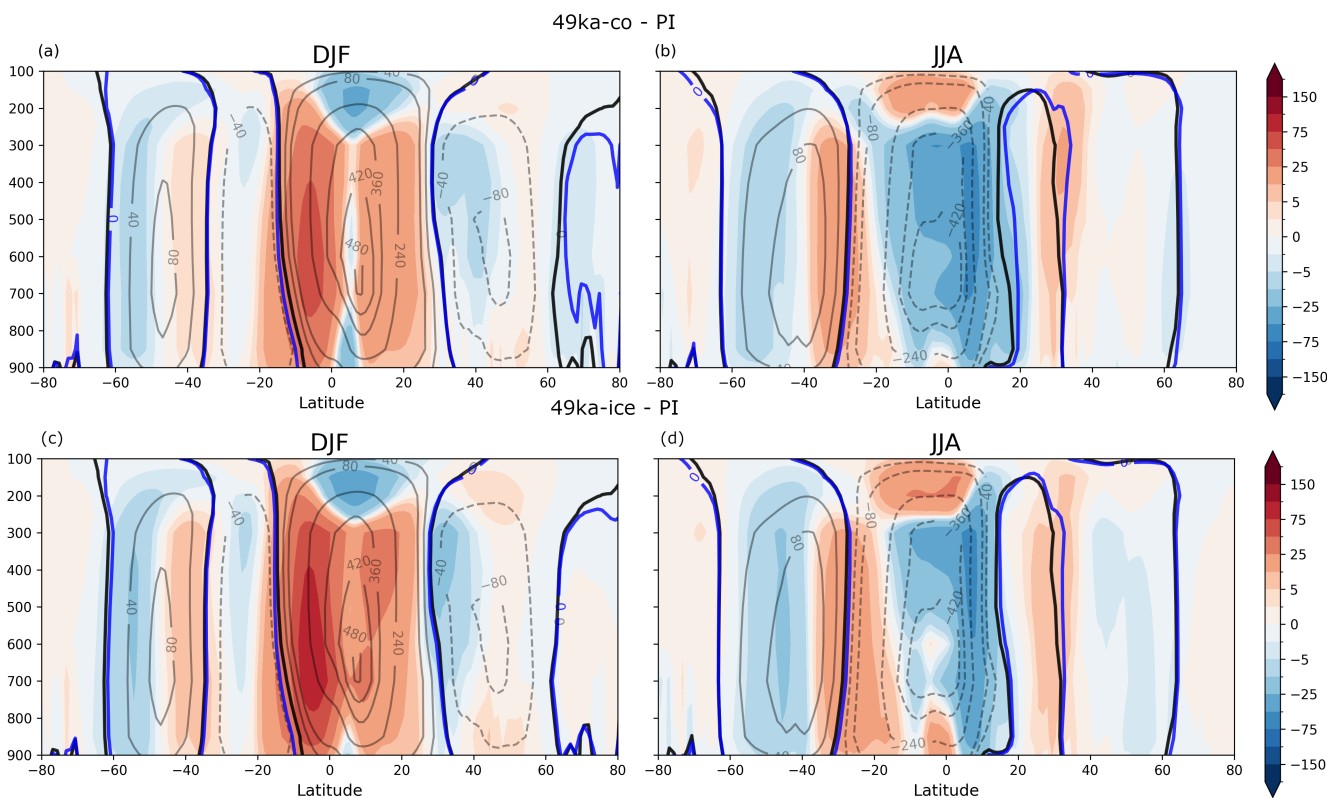

**Figure A6.** Top:Atmospheric mass streamfunction (Sv) anomalies between 49ka-co and PI (colors) for (a) DJF and (b) JJA. Contours are PI (black) absolute streamfunction values. The thick black and blue lines represent the zero contour for PI and 49ka-co, respectively. Bottom: Same as top but for 49ka-ice compared to PI.



*Author contributions.* HS, LM, JRB, and RND designed the study. HS performed the modelling simulations with assistance from DKH. HS conducted the analysis, interpreted the results and wrote the manuscript with input from LM. JRB, RND, DKH and YD provided valuable comments on the manuscript.

*Competing interests.* At least one of the (co-)authors is a member of the editorial board of Climate of the Past

*Acknowledgements.* Himadri Saini, Josephine R. Brown, Russell N. Drysdale, Yanxuan Du, and Laurie Menviel would like to acknowl-
edge funding from the Australian Research Council (ARC) grant DP220102134. Laurie Menviel acknowledges support from ARC grant SR200100008. David K. Hutchinson acknowledges support from ARC grant DE220100279. Josephine R. Brown received support from ARC Centre of Excellence for Weather of the 21st Century (CE230100012). This research was supported by the Australian Government's National Collaborative Research Infrastructure Strategy (NCRIS), with access to computational resources provided by the National Computational Infrastructure (NCI through the National Computational Merit Allocation Scheme.



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
