# Peer review of "The Influence of Glacial Northern Hemisphere Ice Sheets on Atmospheric Circulation"

_EGUsphere, 2025_

## Referee Comment (RC1)

**Saini et al. "The Influence of Glacial Northern Hemisphere Ice Sheets on Atmospheric Circulation"**

**General comments:**

The authors describe the impacts of the Northern Hemisphere (NH) ice sheets at 49,000 years ago on the global atmospheric circulation. They shifted the NH westerlies southward, enhancing Eurasian summer rainfall but reducing it in winter. Their influence also extended to the tropics and the Southern Hemisphere (SH), displacing the Intertropical Convergence Zone (ITCZ) southward and intensifying Australian rainfall. The Laurentide and Antarctic ice sheets further modified the SH circulation, pushing the SH Hadley cell and westerlies equatorward. These results highlight the complex, far-reaching impacts of ice sheets on global climate patterns.

The manuscript is well-written, with clear articulation of the main findings and their broader implications. However, the study's conclusions are highly model-dependent, and further validation is needed to assess whether the simulated responses accurately reflect the glacial climate conditions of 49,000 years ago. Geographically extensive observational data for this specific period would be ideal for evaluation. If such data are unavailable, conducting a Last Glacial Maximum (LGM) experiment using the Australian Earth System Model (ACCESS-ESM1.5) could help assess the impacts of continental ice sheets on glacial climate. Given the computational cost and time constraints, referencing prior studies that used the same model for LGM simulations— along with their data-model comparisons—may suffice as an alternative.

Additionally, the authors could consider compiling a global footprint table for the 49,000-year (or MIS3) climate responses, similar to the Dansgaard-Oeschger (DO) event table in Izumi et al. (2023, QSR). While such a table would not provide quantitative metrics, it would facilitate a qualitative assessment of spatial patterns and enhance model evaluation.

If the authors maintain that this study exclusively examines the model's responses, I recommend publication pending the resolution of minor revisions outlined in the specific comments below.

.

**Specific comments:**

- The rationale for focusing on continental ice sheets at 49 ka rather than the Last Glacial Maximum (LGM) remains unclear. Given the greater availability of paleoclimate records for the LGM, as well as the more extensive ice sheet coverage and its pronounced influence on atmospheric circulation, the choice of 49 ka warrants further justification. The authors should explicitly address why the LGM was deemed unsuitable for this study or why the 49 ka timeframe provides critical insights that the LGM cannot.

- Do the ice sheets of the Last Glacial Maximum (LGM) and Marine Isotope Stage 3 (MIS3 or 49 ka) produce similar large-scale atmospheric and oceanic circulation responses— such as shifts in the Hadley Cell,  Intertropical Convergence Zone (ITCZ), and the Atlantic Meridional Overturning Circulation (AMOC) —even if their magnitudes differ? For

instance, do they induce the same direction of change (e.g., positive or negative anomalies) despite amplitude variations?

- L43-45: Is it possible to take into account millennial-scale fluctuations in this study design? Do the author's results indicate stadial or interstadial climates? Isn't the concentration of research in the northern hemisphere due to the distribution of data for comparison?

- What were the thicknesses of the continental ice sheets—particularly the Laurentide and Antarctic ice sheets—at 49 ka (thousand years ago)?

- L97-98: "The final experiment… 49ka-full…was run for 292 years." Is it true? It does not match the contents of Table 2, if I understand correctly.

- L100: The authors should start a new paragraph here.

- L116: The study suggests that higher sea surface temperatures (SSTs) in the Labrador region are associated with a stronger AMOC in the 49ka-full experiment. Is this AMOC response mechanistically plausible?

- Precision in AMOC Comparisons (L245–246): "AMOC strength at the onset of interstadials versus the pre-industrial (PI) control", or "AMOC strength during stadial versus interstadial periods?"

- In section 3.3, I don't quite understand the relationship between the strength of the Hadley cell and the migration of the ITCZ through the texts and figures. Does the strength of the Hadley cells affect their width?

---

## Referee Comment (RC2)

**Review on**
**Himadri Saini et al.: „The Influence of Glacial Northern Hemisphere Ice Sheets on Atmospheric Circulation"**

**General comments**

The current study is using the Earth system model ACCESS-ESM1.5 to analyze the impact of boundary conditions from 49 kiloyears (ka) before present on the simulated climate with a particular focus on the atmospheric circulation.

The major new aspect is the focus on 49 ka from a modelling perspective and on the role of individual 49 ka boundary conditions on climate. The methods used are standard basic climate model diagnostics. The authors present a solid piece of work with interesting results which warrant publication. However, the presentation of many detailed aspects of the study still requires the specific comments listed below to be addressed before.

**Specific comments**

*Please note that the comments are not sorted by importance but largely follow the structure of the manuscript.*

- In contrast to the authors' statement (1st sentence of abstract), there is already quite some literature published on various aspects of MIS3 climate which I would recommend to reference and to modify the Introduction accordingly. The focus of the Introduction should be more on MIS3 impacts rather than LGM. Malmierca-Vallet et al. (doi:10.5194/cp-19-915-2023), Brandefelt et al. (10.5194/cp-7-649-2011), Zhang et al. (doi:10.1002/2014GL060321), Merkel et al. (doi:10.1016/j.quascirev.2009.11.006) Zhang et al. 2023 (doi:10.1029/2023JD038521), Guo et al. (doi:10.5194/cp-15-1133-2019).

- Abstract l. 8 „Additionally...": The abstract should be slightly modified to make the response to the various combinations of boundary conditions clear.

- The timeseries in Fig. A1 show surface variables only. How about the trends in deeper layers? My guess would be that the ocean temperature and AMOC might not have equilibrated yet which would clearly affect the discussion in section 3.1 (p. 6, ll. 116-130).

- Has this model been applied to other paleo timeslices, e.g. the classical PMIP timeslices? In particular, is there an LGM simulation available from the same model version? That could also provide some insights into the effects of ice sheets on the atmospheric circulation.

- Does this study use exactly the same model setup as for CMIP or are there any paleo adaptations (beyond the application of the paleo boundary conditions)?

- p. 1 l. 21 This needs a reference.

- p. 2: Already very early works by Manabe and Broccoli (1985) and Broccoli and Manabe (1987) provided evidence for a southward displacement of the jet.

- Introduction last sentence: Human migration and settlement patterns are not really tackled in the manuscript. You might rather put this or a similar sentence as an outlook/perspective at the very end of the manuscript.

- Has the stepwise experimental setup been used elsewhere? Then it should be referenced.

- Exp. setup: "PI derived from" - Does this mean that your PI run has been restarted from the Ziehn et al. 2020 simulation and ran for another 1000 years?

- Exp. setup p. 3 l. 83 and Tab. 2: Setting the ice to 52.5 ka is a bit inconsistent/confusing when centering everything around 49 ka. How different are the 52.5 and 49 ka ice sheets taken from Gowan et al. (2021)? A similar question would refer to vegetation (p. 4 l 88).

- It would be nice to have either a figure and/or some sentence briefly describing the characteristics of the 49 ka ice sheet e.g. with respect to modern or LGM in terms of height/extent. Please state clearly in section 2.1.1. which ice sheets you implement (NH and SH), and how different the AIS is between 49 ka and

PI (height, extent, negligible?). The only information you provide comes at a late stage on p.15 l. 273.

- Section 2.1.1: You should explicitly state that the model has PFTs which remain constant throughout each individual experiment and that you modified the PFT distributions for some of your paleo simulations.

- Tab. 2 "Years analysed" - These are 51 or 101 years, respectively, in contrast to what the text on p. 4 says (last 50 or 100 years). Also, l. 98 mentions 292 years, but in Tab. 2 49ka-full seems to cover years 824-1555. Please double-check! What is the reason for averaging over different periods for the analysis (last 50 / last 100 years) Why did you not chose the last years of 49ka-ice for analysis if 49ka-ice runs up to year 824? The choice of 519-569 is unclear.

- I find the experiment name 49ka-ice a bit misleading. To me, it suggests that ice-sheet topography has been implemented in this exper., but you only use 49 ka albedo/vegetation. How about using "49ka-alb" instead?

- According to which criteria did you implement the additional boundary conditions at a particular model year? It seems to be a bit subjective.

- Section 3.1: How about providing a little summary table for the global mean / hemispheric diagnostics for all experiments and reference to that instead of to Fig. 2 which does not explicitly show these numbers?

- Fig. 2: Why is the significance testing only done for precipitation, and not for SAT/SST?

- Section 3.2: When discussing geopotential height (anomalies), I would also write this accordingly and not use the term "pressure" and/or insert "not shown" where required. Readers would look for corresponding (sea level) pressure figures which are not shown.

- Section 3.2: For the jet stream, I would analyze upper tropospheric wind patterns such as uwind at 200 hPa. Wouldn't the Rossby wave response you mention be more evident e.g. at 500 hPa? In contrast to the statement on p. 8 l 174/175, I cannot see the planetary wave structure too clearly.

- For the discussion in section 3.2 (p.7), Fig. A4 is quite clear and helpful. I would recommend to include it into the main part of the manuscript.

- Sections 3.2 to 3.4: I have some concerns regarding the section titles and the discussions in the respective sections. I think that the wording has to be very precise to clearly distinguish between 1) the response to all 49 ka boundary conditions (b.c.) in the 49ka-full experiment and 2) the response in those experiments where only some b.c. have been prescribed. In that sense the section titles "Impact of the ice sheet topography" are partly incorrect unless you explicitly discuss the difference between 49ka-full and 49ka-ice, and even this comparison does not allow to attribute the changes you see exclusively to the ice sheets since you also modify salinity and the land-sea mask etc. The sentence p7 l. 153/154 is an example where to my opinion several aspects are mixed (land-sea distribution, ice-sheet height changes,...). The title of section 3.4 also needs some modification into e.g. „Impact of 49 ka boundary conditions on..."

- Section 3.2: I find it a bit hard to stay on track when following all the details between different seasons, different experiments, and different variables. Maybe starting the paragraph on p. 8 l. 158 with "In the DJF season" could create a clear structure and nicely contrast it with the paragraph starting in l. 169. This also holds for section 3.3 - try to make the structure immediately obvious for the reader (You do it nicely in section 3.4.).

- Section 3.2 and p.15 l. 271/272: moisture (flux convergence), specific humidity: Have you actually looked into these model results? Then it would be good to add "not shown" where appropriate. Otherwise it might be a bit speculative.

- p. 8 l. 164: I would also mention the strong cooling which seems to stand out in response to the albedo change.

- Please be very precise with specifying where changes are happening, e.g. p. 8 l. 175/176 "larger temperature and precip. anomalies", or p. 8 l. 179. "by altered thermal and pressure gradients" is very vague.

- p. 8 l 174: "topography and height" - Isn't height included in topography?

- p. 8 l. 182 and 183: I am confused by the 20 Sv and cannot find this in the shaded part of Fig. 7a (dark reddish colors between about 15S and 10 N). Isn't the anomaly much larger according to the shading? (similar for the 11 Sv anomaly)

- p. 8 l 184 Can you please include a sentence or reference for the link/mechanism between insolation and the Hadley cell strength?

- When comparing the Hadley cell among the different experiments, it would be helpful to have it in the main text and not in the supplement to allow direct comparison for the reader. The authors want to emphasize the role of the different boundary conditions, so it would be helpful to combine figures 7 and A6.

- p. 10 l. 190: "temperature gradient between the two hemispheres" - How about quantifiying this for all experiments and add it to the diagnostics table suggested above?

- p. 10/11 l. 197/198: I suggest to rephrase this sentence. => "Temperature contrasts between hemispheres are amplified due to the introduction of ice sheet topography which induces localized warming in Siberia and strong Antarctic cooling, but also due to the replacement of ocean grid cells by land." It might also make things clearer if you explicitly specify the direction of the Hadley cell shift in relation to the hemisphere temperature contrast (shift towards the warmer hemisphere?)

- p. 11: Since you already include the SH westerlies into Fig. 4, you could move the paragraph on SH westerlies to the end of section 3.2, discuss them there and keep a "tropical"-only focus in section 3.3.

- section 3.4 p. 12 ll. 208/209: I do not fully agree with this statement. Doesn't Fig. 5c (49ka-co) already show the increase in Australian rainfall? You mention the strong obliquity change at 49 ka, so you might need to take this into account as well and not attribute everything to the ice sheets. I would start the section with l. 209 (slightly modified) and carefully phrase the following part in the light of the different boundary condtions, not just ice sheets, especially when referring to the 49ka-full figures (3,7) in the subsequent paragraphs. You could add for instance "in response to the 49 ka boundary conditions" to the sentence in p. 13 l. 223.

- p. 14: AMOC strength during MIS3 seems to vary a lot among timeslices and models. A CCSM3 35 ka simulation initialized from LGM showed a weak AMOC (Merkel et al. doi:10.1016/j.quascirev.2009.11.006).

- p. 15 l. 275: What do you mean by "more dynamic"? The zonal response in Löfverström' study is also a dynamic one, isn't it?

- p. 15 l. 279/280: There are studies which demonstrate the impact of glacial boundary conditions beyond the N. Atlantic sector (see doi indications above, but also for instance DiNezio et al. doi:10.1126/sciadv.aat9658, Mohtadi et al. doi:10.1038/nature13196, Shi et al. 10.5194/cp-19-2157-2023).

- p. 15 l. 282: => "compared to our simulations"

- p. 15 vs. section 3: Some parts of the discussion on p. 15 have already been raised in section 3. Section 4 reads quite well, so you might consider to shorten section 3 and leave the interpretation for section 4.

- section 4 Fig. 8: This is a nice summary which might already be worth to refer to during section 3. When averaging over the North Atlantic, is this all ocean grid points in a latitude range or did you chose some lat-lon box?

- section 4: I would recommend to modify the titles in bold of this section. I don't think these subsections can be clearly separated, and as noted above, phrasing should be done very carefully and with less focus on the ice sheets in the titles due to the experimental setup. It is appropriate to discuss the important role of the ice sheets, but the titles should be more general.

- p. 17 l.309: You might want to be a little bit more specific about the term "water availability" (soil moisture, atmospheric moisture content,...?).

- p. 17 l. 313: Wouldn't it make sense to also refer to your Fig. 3g to support your argument related to the insolation changes?

- The Conclusions section is very short and does not mention MIS3 at all. It should be slightly rephrased in order to align better with the motivation and the main focus of the manuscript. Furthermore, the phrasing of the impacts of the ice sheets suggests that separate experiments have been conducted to isolate the respective impact of NH and SH ice sheets (LIS, AIS). Please make the wording more concise.

- Data availability: Will also the code, e.g. to calculate the atmospheric mass streamfunction, be made available? Which density has been used for the calculation in Sv?

- To my knowledge, in a student's t-test, you would call 95% the confidence level and 5% the significance level. Please correct the corresponding text and figure captions.

- Fig. A1 caption mentions „simulated ice sheet", but according to the Methods section, ice sheets are pre-scribed as constant forcing in this study.

- Fig. A3 mentions „North Atlantic gyre strength". Is it the barotropic streamfunction?

**Technical corrections**

- Please double-check all acronyms, not all of them have been explained or explained where they appear for the first time (e.g. CMIP6 p. 3 l. 63, CASA-CNP p. 3 l. 67)

- p.  1 l. 3 "simulate a glacial climate" => "simulate the glacial climate"

- p. 1 l. 14 "by their extent" => "by the ice sheet extent" (and also height?)

- p. 1 l.19 "on the atmospheric"

- p. 2 l. 45 Menviel et al., 2020 => Menviel et al., 2020 and references therein

- Section 2.1 l. 67: "uses" => "consists in"

- p. 3 l. 81/82: => "with only the orbital parameters... and GHG concentrations ... being set to 49 ka va-lues..."

- p. 4: Tab. 1: I would just put 49 ka instead of 49ka-full.

- p. 6 l. 1 => "divides the tropical precipitation amount"

- p. 6 l. 4 "within the same range" => „over the same vertical/pressure range". Is there a reference for cho-sing 400-600 hPa?

- Tab. 2: caption: "in all" => "in the different 49 ka experiments"; "in the full time-series shown in" => "as shown in the timeseries of Fig. A1"

- Fig. 1 caption: a), b) missing; "per grid cell" => "of each grid cell"

- Fig. 2: Sea-ice edges are very hard to see in Fig. 2b. How about chosing polar stereographic projection instead?

- Fig. 2 caption: "ocean ice" => "sea ice". "Contours are" => "Contours show". I would also start the capti-on with "Annual mean anomalies of..."

- l. 125: => "by 5 to 6° latitude"

- l. 127: NADW => Do you miss "formation" here?

- p. 7 l. 137: I guess you are referring to the meridional temperature gradient here. Please add.

- p. 7 l. 138: => "shift by ~6°"

- Fig. 3 caption: I would start with the caption with "Anomalies between 49ka-full and PI for (a,b) SAT (°C)..."

- Fig. 3c-f: What happens at 0°E in the vector plots? Have you also looked at higher levels to avoid the plotting conflict with topography?

- p. 14 l. 269: => "landmasses"

- Fig. 8: Since you have a), b) in the Figure, you could adjust the caption accordingly. You could also make the lat-lon specifications in brackets consistent (hyphen, colon) and make the Figure and caption consis-tent (Europe/Eurasia).

- Fig. A1: The caption does not mention the lower panels and does not have references to "left panels", "middle panels" etc. You could also add a), b), labels. Please also mention what is shown, I guess it should read "Timeseries of anomalies of annual mean surface air temperature...".

- Fig. A2: What exactly is shown here? Have you taken the SW downward radiation at the top of the atmosphere/model?

- Fig. A3: Please mention in the caption that the white areas mark grid cells with continental ice, and that for the 49 ka simulation, continental outlines are shown based on the adjusted land-sea mask. I would also write "...the 49ka_full to PI difference".

- Fig. A5: caption incomplete: It is missing vector descriptions.

---

## Author Comment (AC1)

**Saini et al. "The Influence of Glacial Northern Hemisphere Ice Sheets on Atmospheric Circulation"**

**General comments:**

The authors describe the impacts of the Northern Hemisphere (NH) ice sheets at 49,000 years ago on the global atmospheric circulation. They shifted the NH westerlies southward, enhancing Eurasian summer rainfall but reducing it in winter. Their influence also extended to the tropics and the Southern Hemisphere (SH), displacing the Intertropical Convergence Zone (ITCZ) southward and intensifying Australian rainfall. The Laurentide and Antarctic ice sheets further modified the SH circulation, pushing the SH Hadley cell and westerlies equatorward. These results highlight the complex, far-reaching impacts of ice sheets on global climate patterns.

The manuscript is well-written, with clear articulation of the main findings and their broader implications. However, the study's conclusions are highly model-dependent, and further validation is needed to assess whether the simulated responses accurately reflect the glacial climate conditions of 49,000 years ago. Geographically extensive observational data for this specific period would be ideal for evaluation. If such data are unavailable, conducting a Last Glacial Maximum (LGM) experiment using the Australian Earth System Model (ACCESS-ESM1.5) could help assess the impacts of continental ice sheets on glacial climate. Given the computational cost and time constraints, referencing prior studies that used the same model for LGM simulations—along with their data-model comparisons—may suffice as an alternative.

Additionally, the authors could consider compiling a global footprint table for the 49,000-year (or MIS3) climate responses, similar to the Dansgaard-Oeschger (DO) event table in Izumi et al. (2023, QSR). While such a table would not provide quantitative metrics, it would facilitate a qualitative assessment of spatial patterns and enhance model evaluation.

If the authors maintain that this study exclusively examines the model's responses, I recommend publication pending the resolution of minor revisions outlined in the specific comments below.

***"Please find the Author's response in blue and the modified text in the manuscript in green."***

We thank the reviewer for this thoughtful comment and agree to state that the results shown in this study are highly model-dependent. At present, the 49 ka simulation is the only glacial time slice available for ACCESS-ESM1.5. We are currently in the process of simulating the LGM using the same model, following the PMIP4 protocol. This model has previously been used to simulate the last interglacial (lig127k) time slice as part of the Tier 1 PMIP4-CMIP6 experiments (Yeung et al., 2021: https://doi.org/10.5194/cp-17-869-2021, 2021.), MIS9e (~336-321 ka; Duboc et al., 2025, https://cp.copernicus.org/articles/21/1093/2025/), and mid-Holocene (6 ka; Mackallah et al., 2022, https://www.publish.csiro.au/ES/ES21031). This information is now added in section 2.1.

As noted, the computational cost of running this comprehensive Earth system model is considerable, ~16 model years per day.

Nevertheless, we have now included some model–data comparisons for SAT and SST. We compare our simulated results for 49 ka relative to PI with the proxy records, focusing on Greenland Interstadial 13 (GI13; ~49 ka) versus PI (Fig. 2 in the revised manuscript).

[Figure]

Fig. R1: Annual mean anomalies of surface air temperature (SAT), sea surface temperature (SST), and precipitation for 49 ka climate (exp 49ka-full) compared to PI. Contours show seasonal mean 15% sea ice concentration for (orange) 49 ka and (black) PI during DJF (dashed) and JJA (solid). Stippling indicates non-significant changes based on a Student's t-test at 95% confidence level. Stars in subpanels (a) and (b) indicate qualitative estimates of the changes between Greenland Interstadial (GI 13) and PI climate, as inferred from the proxy records (MDO1-2444,ODP-977A,MD95-2043,MD01-2443,MD95-2042 [Martrat et al., 2007]; ODP-1233 [Kaiser et al., 2005]; TR163-19 [Lea et al., 2000]; RC13-110 [Feldber and Mix, 2003]; ODP846B [Martinez et al., 2003]; TG-7 [Calvo et al., 2001]; MD97-2120 [Pahnke et al., 2003]; MD07-3128 [Caniupan et al., 2011]; EPICA Dronning Maud Land [EPICA community members]; WAIS [WAIS community members]; NGRIP [Huber et al., 2006, Kindler et al., 2014]; SO188-17286-1 [Lauterbach et al., 2020]. Dark (light) blue colors represent much colder (slightly colder) climate).

The following text is also added in the manuscript.

Lines 146-147:

"The simulated SATs and SSTs anomalies show a good agreement with proxy records indicating that the simulation presents a qualitatively appropriate estimate of the changes between Greenland Interstadial (GI 13; ~49ka-50ka) and PI climate (Fig. 2a,b)."

Our focus on 49 ka at this stage, rather than the LGM, is motivated by a broader project investigating millennial-scale variability in Australian hydroclimate. Specifically, we aim to simulate Heinrich Event 5 (H5), which occurred around 49 ka. This paper represents the first in a series of studies and provides a baseline description of the glacial climate at 49 ka prior to H5. In forthcoming work, we will present H5 model simulations and compare them with new speleothem records from Australia that span this interval. This will allow us to directly evaluate the modelled hydroclimate response against proxy data.

**Specific comments:**

• The rationale for focusing on continental ice sheets at 49 ka rather than the Last Glacial Maximum (LGM) remains unclear. Given the greater availability of paleoclimate records for the LGM, as well as the more extensive ice sheet coverage and its pronounced influence on atmospheric circulation, the choice of 49 ka warrants further justification. The authors should explicitly address why the LGM was deemed unsuitable for this study or why the 49 ka timeframe provides critical insights that the LGM cannot.

We thank the reviewer for this comment. We agree that the LGM is an interesting time period with greater availability of paleoclimate records, however, MIS3 is a period of high interest because of the occurrence of millennial-scale variability. Some of this millennial-scale variability could have arisen from the interaction between ice-sheets and climate, thus motivating studies on that time period. This time period also provides information on the impact of a mid-size Northern Hemispheric ice-sheet on climate.

Currently, Lines 29-29 describe previous studies that focus on LGM to understand the impact of ice sheets. Lines 40-52 further elaborate on the motivation to investigate MIS3. While significant progress has been made in understanding LGM climate, the influence of ice sheets during the MIS3 period—when millennial-scale climate variability was pronounced—remains less well understood.

In the revision, we have now included a few modelling studies which have assessed the impact of MIS3 boundary conditions.

Lines 47-57:

"In contrast, the climate of MIS 3 (65–25 ka) has received comparatively less attention in the context of ice sheet–atmosphere interactions. Existing studies of MIS 3 have largely concentrated on large-scale oceanic responses, such as AMOC variability, surface temperature patterns, and sea ice changes (Malmierca-Vallet and Sime, 2023; Brandefelt et al., 2011; Merkel et al., 2010; Guo et al., 2019). Some have investigated tropical climate variability, including changes in ENSO behaviour during this period (Brandefelt et al., 2011; Merkel et al., 2010), while others briefly mention reduced global precipitation, a southward-shifted ITCZ, intensified Southern Hemisphere (SH) westerly wind stress, and strengthened NH trade winds alongside weakened SH trade winds (Guo et al., 2019). However, the detailed, mechanistic impacts of NH ice sheets on atmospheric circulation during MIS 3—particularly beyond the North Atlantic—remain underexplored. This gap may be partly due to the significant millennial-scale climate variability that characterizes MIS 3—including abrupt temperature fluctuations in Greenland of 8–10°C (Huber et al., 2006) and multiple episodes of AMOC weakening (Menviel et al. (2020) and references therein)—which make it challenging to isolate equilibrium climate response to boundary conditions."

We selected 49 ka in particular due to its unique combination of boundary conditions: it features relatively high obliquity—higher than both the PI and the Last Interglacial—alongside low atmospheric $CO_2$ concentrations (~200 ppm), similar to the LGM.

• Do the ice sheets of the Last Glacial Maximum (LGM) and Marine Isotope Stage 3 (MIS3 or 49 ka) produce similar large-scale atmospheric and oceanic circulation responses—such as shifts in the Hadley Cell, Intertropical Convergence Zone (ITCZ), and the Atlantic Meridional Overturning Circulation (AMOC) —even if their magnitudes differ? For instance, do they induce the same direction of change (e.g., positive or negative anomalies) despite amplitude variations?

As far as we know, the atmospheric circulation response to LGM and MIS3 ice-sheets has not been studied in detail within a similar modelling framework. Our results thus present important information related to the impact of MIS3 ice-sheets on atmospheric circulation. Further studies should assess the impact of growing glacial ice-sheets on large-scale atmospheric circulation.

In contrast, AMOC changes have been more widely studied. Our results show that the influence of ice sheet topography on AMOC strength is indeed similar between the LGM and 49 ka ice sheet configurations in some studies (see Lines 248–254). However, we have also noted large variations among different MIS3 time slices and models, as mentioned in the new text below.

Lines 290-294:

Model simulations also show considerable spread in AMOC strength across different MIS 3 timeslices and model setups. For instance, a Community Climate System Model version 3 (CCSM3) simulation at 35 ka, initialized from the LGM, showed a ~40% reduction in AMOC strength (Merkel et la., 2010), while the same model produced a ~50% reduction at 44 ka, also initialised from the LGM state (Brandefelt et al., 2011). In contrast, the Norwegian Earth System Model (NorESM) simulations indicate a ~13% strengthening of the AMOC at 38 ka (Guo et al., 2019). The simulated AMOC in our model is strong at 49 ka (31 Sv) with a ~47\% increase relative to PI.

Additionally, the response of the North Atlantic westerlies to ice sheets is also consistent between the two climates (see Lines 264–267). Thus, while the magnitude of the climate response may differ, some of the directional changes are comparable across the two glacial periods. Nevertheless, previous studies (Zhang et al., and Armstrong et al., 2022) have shown that even relatively small changes in NH ice-sheets can have significant impacts. Therefore, while our study provides an estimate of MIS3 ice-sheet impacts on climate, additional work is required to fully understand how glacial ice-sheets influenced large scale circulation.

• L43-45: Is it possible to take into account millennial-scale fluctuations in this study design?

Yes, a companion study is currently in preparation that will explore millennial-scale fluctuations in AMOC during Heinrich Stadial 5 (H5). This follow-up study builds upon the 49 ka control experiment presented here and will specifically address the transient response of the climate system to millennial-scale perturbations.

Do the author's results indicate stadial or interstadial climates?

Our results resemble an interstadial climate as the AMOC is stronger than PI. We have now clearly mentioned this in the manuscript.

Line 278

"Here, the simulated state in 49ka-full experiment represents an interstadial climate."

Isn't the concentration of research in the northern hemisphere due to the distribution of data for comparison?

Yes, the concentration of research in the North Atlantic region is likely influenced by the greater availability of paleoclimate data from that area, as well as the presence of extensive ice sheets over North America, which had a strong impact on regional and global climate. This makes our results particularly interesting, as they highlight that the influence of glacial ice sheets extends well beyond the North Atlantic, with significant impacts in other regions, including the Southern Hemisphere

We have now added this to our discussion.

Lines 273-277

"Additionally, much of the existing research has focused primarily on the North Atlantic region, driven by the greater availability of paleoclimate data from this area and the presence of extensive North American ice sheets that strongly influenced both regional and global climate. Our results add to this literature by demonstrating that the influence of glacial ice sheets extends well beyond the North Atlantic, with significant climatic impacts also occurring in other regions, including the SH."

• What were the thicknesses of the continental ice sheets—particularly the Laurentide and Antarctic ice sheets—at 49 ka (thousand years ago)?

The ice thickness of the Laurentide Ice sheet was more than 3km, while the Antarctic ice sheet reached an altitude of more than 3.6km. The height as well as the extent at this time period were less than LGM. We have now included this information in the manuscript and the below plot of ice sheet thickness in Fig. 1.

[Figure]

Fig.R2: Ice sheet thickness and extent from Gowan et al., (2021) for PI, 49 ka (represented by 52.5 ka as described in the manuscript), and LGM climates.

Lines 101-108:

"The NH ice sheets at 49 ka include two separate ice sheets over the North American continent—Laurentide in the east and Cordilleran in the west—as well as the Greenland Ice Sheet and Fennoscandian and Eurasian ice sheets (Figure 1b). Their extent and height are smaller than during the LGM, which featured a single, extensive LIS over North America and continuous ice sheet coverage from Eurasia to Scandinavia, fully covering the Barents Sea. The Antarctic Ice Sheet (AIS) at 49 ka also has a smaller extent compared to that of the LGM, especially over the Ross and Weddell Seas. At 49 ka, the LIS was up to 3 km thick, while the AIS was up to 3.6 km thick. In the present day, permanent ice cover in the Northern Hemisphere is restricted to Greenland (Figure 1a)."

• L97-98: "The final experiment... 49ka-full...was run for 292 years." Is it true? It does not match the contents of Table 2, if I understand correctly.

Apologies for this confusion. We have corrected it to 760 years now.

• L100: The authors should start a new paragraph here.

Done.

• L116: The study suggests that higher sea surface temperatures (SSTs) in the Labrador region are associated with a stronger AMOC in the 49ka-full experiment. Is this AMOC response mechanistically plausible?

Yes, this response is mechanistically plausible as the AMOC strengthening enhances the oceanic heat transport towards the Labrador Sea. However, due to the strong heat loss to the atmosphere, surface waters become cold enough to form deep waters.

• Precision in AMOC Comparisons (L245–246): "AMOC strength at the onset of interstadials versus the pre-industrial (PI) control", or "AMOC strength during stadial versus interstadial periods?"

Rephrased to the suggested change.

• In section 3.3, I don't quite understand the relationship between the strength of the Hadley cell and the migration of the ITCZ through the texts and figures. Does the strength of the Hadley cells affect their width?

In general, changes in the strength of the Hadley circulation can be associated with changes in its width and the latitude of the ITCZ, but these relationships are not always linear or straightforward. In our study, we observe that an intensification of the NH Hadley cell during DJF is accompanied by an equatorward shift of its ascending branch (i.e., the ITCZ). This suggests that the Hadley cell is not only strengthening but also narrowing.

---

## Author Comment (AC2)

**Review on**
**Himadri Saini et al.: „The Influence of Glacial Northern Hemisphere Ice Sheets on Atmospheric Circulation"**

**General comments**

The current study is using the Earth system model ACCESS-ESM1.5 to analyze the impact of boundary con- ditions from 49 kiloyears (ka) before present on the simulated climate with a particular focus on the atmo- spheric circulation.

The major new aspect is the focus on 49 ka from a modelling perspective and on the role of individual 49 ka boundary conditions on climate. The methods used are standard basic climate model diagnostics. The aut- hors present a solid piece of work with interesting results which warrant publication. However, the presentati- on of many detailed aspects of the study still requires the specific comments listed below to be addressed before.

We thank the reviewer for their very thorough and constructive comments which has really led to the improvement of this manuscript and research quality. Please see the point-by-point answers to each of the comments below.

*Author comments are provided in "blue" and new added text is provided in green.*

**Specific comments**

*Please note that the comments are not sorted by importance but largely follow the structure of the manuscript.*

- In contrast to the authors' statement (1st sentence of abstract), there is already quite some literature pu- blished on various aspects of MIS3 climate which I would recommend to reference and to modify the In- troduction accordingly. The focus of the Introduction should be more on MIS3 impacts rather than LGM. Malmierca-Vallet et al. (doi:10.5194/cp-19-915-2023), Brandefelt et al. (10.5194/cp-7-649-2011), Zhang et al. (doi:10.1002/2014GL060321), Merkel et al. (doi:10.1016/j.quascirev.2009.11.006) Zhang et al. 2023 (doi:10.1029/2023JD038521), Guo et al. (doi:10.5194/cp-15-1133-2019).

  We thank the reviewer for this valuable suggestion. We agree these studies have investigated MIS 3 climate, including the role of ice sheets, ocean circulation, and variability patterns such as ENSO and AMOC. We have revised both the abstract and introduction to reflect this literature more accurately.

  Specifically, we now reference and briefly discuss Malmierca-Vallet et al. (2023) and Guo et al. (2019), both of which explore aspects of atmospheric and oceanic circulation beyond the North Atlantic. We also cite Brandefelt et al. (2011) and Merkel et al. (2010) for their contributions to understanding MIS 3 climate state and variability, including the role of AMOC and ENSO. However, we chose not to include Zhang et al. (2023) in the revised text because their study focuses on the influence of AMOC variability on global monsoon precipitation during MIS 3, rather than the direct influence of ice sheets themselves.

  Lines 47-57:

  "In contrast, the climate of MIS 3 (65–25 ka) has received comparatively less attention in the context of ice sheet–atmosphere interactions. Existing studies of MIS 3 have largely concentrated on large-scale oceanic responses, such as AMOC variability, surface temperature patterns, and sea ice changes (Malmierca-Vallet and Sime, 2023; Brandefelt et al., 2011; Merkel et al., 2010; Guo et al., 2019). Some have investigated tropical climate variability, including changes in ENSO behaviour during this period (Brandefelt et al., 2011; Merkel et al., 2010), while others briefly mention reduced global precipitation, a southward-shifted ITCZ, intensified Southern Hemisphere (SH) westerly wind stress, and strengthened NH trade winds alongside weakened SH trade winds (Guo et al., 2019). However, the detailed, mechanistic impacts of NH ice sheets on atmospheric circulation during MIS 3—particularly beyond the North Atlantic—remain underexplored. This gap may be partly due to the significant millennial-scale climate variability that characterizes MIS 3—including abrupt temperature fluctuations in Greenland of 8–10°C (Huber et al., 2006) and multiple episodes of AMOC weakening (Menviel et al. (2020) and references therein)—which make it challenging to isolate equilibrium climate response to boundary conditions."

- Abstract l. 8 „Additionally...": The abstract should be slightly modified to make the response to the various combinations of boundary conditions clear.

  Modified as below.

During the last glacial period, Northern Hemisphere (NH) ice sheets significantly influenced atmospheric circulation, however most studies primarily focused on the North Atlantic. Here, we investigate the influence of ice sheets on global climate during Marine Isotope Stage 3 (MIS 3, 65-25 ka), a period marked by prominent millennial-scale variability. Using the Australian Earth System Model (ACCESS-ESM1.5), we simulate the glacial climate around 49,000 years ago (49 ka). Our findings demonstrate that the NH ice sheets induced a southward shift of the NH westerlies during boreal summer and winter, increasing rainfall over Eurasia during summer but reducing it in winter. The influence of orbital parameters and greenhouse gases alone produces only minor circulation changes. Adding ice sheet albedo intensifies the NH cooling and strengthens the westerlies but does not significantly alter their latitudinal position. The ice sheets' influence also extends across the tropics and the Southern Hemisphere (SH). The most pronounced southward displacements of the Intertropical Convergence Zone (ITCZ) and the NH Hadley cell during austral summer—associated with enhanced Australian rainfall—occur when all ice sheet-related boundary conditions are included. In the Southern Hemisphere, the presence of ice sheets leads to an equatorward shift of the SH Hadley cell during JJA, with the largest displacement occurring when all boundary conditions are combined. Additionally, SH westerlies show no latitudinal shift in response to orbital parameters, greenhouse gases, or albedo alone, but exhibit a pronounced equatorward shift when ice sheet topography is included. These findings highlight the non-linear interactions between ice sheets, large-scale atmospheric circulation, and precipitation patterns.

- The timeseries in Fig. A1 show surface variables only. How about the trends in deeper layers? My guess would be that the ocean temperature and AMOC might not have equilibrated yet which would clearly affect the discussion in section 3.1 (p. 6, ll. 116-130).

  Please find below the time series of anomalies in globally averaged ocean temperature and AMOC. Over the last 100 years, temperature changes by 0.004°C and AMOC changes by 0.23 Sv (Fig. R1 below). This shows both ocean temperature and AMOC are in quasi-equilibrium.

[Figure]

Fig.R1. Time series of anomalies of (left) globally averaged ocean temperature and (right) AMOC for (black) PI, (cyan) 49ka-co, (orange) 49ka-alb, and (blue) 49ka-full compared to PI. 49ka-co is orbital parameters+GHGs, 49ka-alb is 49ka-co with albedo and vegetation changes. Additional changes to 49ka-alb are shown in magenta colors (49ka-alb+). At year 547, surface salinity is gradually increased to account for the 44 m sea level drop relative to PI, based on the prescribed ice sheet, achieving a global average increase of +0.33 psu from PI levels. At year 729, the Bering Strait is closed. These experiments are not analysed in the manuscript. At year 824, the 49 ka topography is implemented. River runoff is adjusted to reflect the 49 ka topography at year 1258. Darker colors indicate the time-averaged period for each experiment used in the analysis. The horizontal red line is the zero line.

- Has this model been applied to other paleo timeslices, e.g. the classical PMIP timeslices? In particular, is there an LGM simulation available from the same model version? That could also provide some insights into the effects of ice sheets on the atmospheric circulation.

  We are currently in the process of simulating the LGM using the same model, following the PMIP4 protocol. This model has previously been used to simulate the last interglacial (lig127k) time slice as part of the Tier 1 PMIP4-CMIP6 experiments (Yeung et al., 2021: https://doi.org/10.5194/cp-17-869-2021, 2021.), MIS9e (~336-321 ka; Duboc et al., 2025, https://cp.copernicus.org/articles/21/1093/2025/), and mid-Holocene (6 ka; Mackallah et al., 2022, https://www.publish.csiro.au/ES/ES21031). This information is now added in section 2.1.

Our focus on 49 ka at this stage, rather than the LGM, is motivated by a broader project investigating millennial-scale variability in Australian hydroclimate. Specifically, we aim to simulate Heinrich Event 5 (H5), which occurred around 49 ka. This paper represents the first in a series of studies and provides a baseline description of the glacial climate at 49 ka prior to H5. In forthcoming work, we will present H5 model simulations and compare them with new speleothem records from Australia that span this interval. This will allow us to directly evaluate the modelled hydroclimate response against proxy data.

- Does this study use exactly the same model setup as for CMIP or are there any paleo adaptations (beyond the application of the paleo boundary conditions)?

  Yes, this study uses the same model configuration as in the CMIP6 experiments, including identical physics schemes, coupling frequency, and resolution. The only differences from the CMIP configuration are the application of paleo boundary conditions appropriate for 49 ka BP (i.e., orbital parameters, greenhouse gas concentrations, ice sheet topography and extent, and sea level).

  No additional model tuning or physics modifications were applied specifically for the paleo context. As such, any differences in the climate response arise solely from the imposed boundary conditions

- p. 1 l. 21 This needs a reference.

  References Added. (Seager et al., 2002 and Brayshaw et al., 2009.)

- p. 2: Already very early works by Manabe and Broccoli (1985) and Broccoli and Manabe (1987) provided evidence for a southward displacement of the jet.

  Apologies for missing these earlier references. They are added now.

- Introduction last sentence: Human migration and settlement patterns are not really tackled in the manuscript. You might rather put this or a similar sentence as an outlook/perspective at the very end of the manuscript.

  This has been removed from the manuscript now, as another study within our project is looking into it.

- Has the stepwise experimental setup been used elsewhere? Then it should be referenced.

  Not to our knowledge. We adopted a stepwise approach because this is the first time such experiments, were being conducted with the ACCESS model. Each experiment was configured progressively as the necessary capabilities became available.

- Exp. setup: "PI derived from" - Does this mean that your PI run has been restarted from the Ziehn et al. 2020 simulation and ran for another 1000 years?

  No, we used the existing 1000-year piControl simulation described by Ziehn et al., and analysed the average of the final 100 years from that run. We have clarified this in the revised manuscript text.

  Lines 91-93:

  "The pre-industrial (PI) simulation is the 1000-year piControl run based on the year 1850 CE as described by Ziehn et al., 2020. We analyse the average of the last 100 years of this simulation (Figure A1}, cyan line). Orbital parameters and GHG concentrations for the PI configuration are provided in Table 1."

- Exp. setup p. 3 l. 83 and Tab. 2: Setting the ice to 52.5 ka is a bit inconsistent/confusing when centering everything around 49 ka. How different are the 52.5 and 49 ka ice sheets taken from Gowan et al. (2021)? A similar question would refer to vegetation (p. 4 l 88).

  Thank you for pointing this out. The Gowan et al. (2021) ice sheet reconstructions are available at 47.5, 50, and 52.5 ka. We selected the **52.5 ka** reconstruction for this study because the associated global sea level estimate is closer to other independent reconstructions at ~49 ka, whereas the 50 ka scenario deviates more significantly. Gowan et al. (2021) also provide two scenarios for each time slice: a **maximum** scenario (with ice cover over Hudson Bay) and a **minimum** scenario (ice-free Hudson Bay). We chose the **maximum scenario at 52.5 ka** to better represent the likely glacial conditions immediately preceding Heinrich Stadial 5 (H5), which is a focus of our broader project. This selection of maximum scenario also in line with the MIS 3 protocol for baseline climate mentioned in Malmierca-Vallet et al. (2023). We acknowledge that there are regional differences between the 50 ka and 52.5 ka ice sheet reconstructions (Fig.R2).

[Figure]

Fig. R2. Anomalies of ice sheet thickness (Gowan et al., 2021) between 52.5 ka and 50 ka.

For vegetation, we refer to the corresponding 52 ka snapshot to ensure consistency with the ice sheet reconstruction. The available time slices for vegetation were 48, 50, and 52 ka. As shown in Fig.R3, differences between the 50 ka and 52 ka vegetation distributions are minor.

[Figure]

Fig. R3. Simulated biomes for 52 ka and 50 ka (Allen et al., 2020).

- It would be nice to have either a figure and/or some sentence briefly describing the characteristics of the 49 ka ice sheet e.g. with respect to modern or LGM in terms of height/extent. Please state clearly in section 2.1.1. which ice sheets you implement (NH and SH), and how different the AIS is between 49 ka and PI (height, extent, negligible?). The only information you provide comes at a late stage on p.15 l. 273.

[Figure]

Fig.R4: Ice sheet thickness and extent from Gowan et al., (2021) for PI, 49 ka (represented by 52.5 ka as described in the manuscript), and LGM climates.

We thank the reviewer for this suggestion. We have added Fig. R4 in Fig. 1, and the below text in the main manuscript describing the characteristics of the 49 ka ice sheets.

Lines 101-108:

"The NH ice sheets at 49 ka include two separate ice sheets over the North American continent—Laurentide in the east and Cordilleran in the west—as well as the Greenland Ice Sheet and Fennoscandian and Eurasian ice sheets

(Figure 1b) Their extent and height are smaller than during the LGM, which featured a single, extensive LIS over North America and continuous ice sheet coverage from Eurasia to Scandinavia, fully covering the Barents Sea (Figure 1c). The Antarctic Ice Sheet (AIS) at 49 ka also has a smaller extent compared to that of the LGM, especially over the Ross and Weddell Seas. At 49 ka, the LIS was up to 3 km thick, while the AIS was up to 3.6 km thick. In the present day, permanent ice cover in the Northern Hemisphere is restricted to Greenland (Figure 1a)."

- Section 2.1.1: You should explicitly state that the model has PFTs which remain constant throughout each individual experiment and that you modified the PFT distributions for some of your paleo simulations.

  Done.  Line 97:

  "The model uses prescribed Plant Functional Types (PFTs) that remain fixed during each individual simulation. We modified the PFT distributions in selected simulations to reflect the altered boundary conditions."

- Tab. 2 "Years analysed" - These are 51 or 101 years, respectively, in contrast to what the text on p. 4 says (last 50 or 100 years). Also, l. 98 mentions 292 years, but in Tab. 2 49ka-full seems to cover years 824-1555. Please double-check! What is the reason for averaging over different periods for the analysis (last 50 / last 100 years) Why did you not chose the last years of 49ka-ice for analysis if 49ka-ice runs up to year 824? The choice of 519-569 is unclear.

  Thank you for pointing this out. We have now clarified the text and table to ensure consistency.

  The 49ka-ice (**now changed to 49ka-alb**) experiment includes orbital, GHG, albedo, and vegetation changes. It starts from the model year 297 and was run for a total of 272 years (from model year 297-569). Since the simulation had not reached equilibrium, we analyzed only the last 50 years for this experiment. 49ka-co and 49ka-full are closer to equilibrium, therefore, we analysed the last 100 years of these runs. We have verified that averaging over 50 instead of 100 years in these simulations does not significantly affect the results.

  We have also corrected that the 49ka-full experiment has been run for a total of 760 years.

[Figure]

We have also modified the fig caption as below:

Fig. R5. Time series of anomalies of annual mean (left) globally, (middle) southern hemisphere (SH), and (right) northern hemisphere (NH) averaged surface air temperature (SAT) and sea surface temperature (SST) for (black) PI, (cyan) 49ka-co, (orange) 49ka-alb, and (blue) 49ka-full compared to PI. 49ka-co is orbital parameters+GHGs, 49ka-alb is 49ka-co with albedo and vegetation changes. Additional changes to 49ka-alb are shown in magenta colors (49ka-alb+). At year 547, surface salinity is gradually increased to account for the 44 m sea level drop relative to PI, based on the prescribed ice sheet, achieving a global average increase of +0.33 psu from PI levels. At year 729, the Bering Strait is closed. These experiments are not analysed in the manuscript. At year 824, the 49 ka topography is implemented. River runoff is adjusted to reflect the 49 ka topography at year 1258. Darker colors indicate the time-averaged period for each experiment used in the analysis. The horizontal red line is the zero line.

- I find the experiment name 49ka-ice a bit misleading. To me, it suggests that ice-sheet topography has been implemented in this exper., but you only use 49 ka albedo/vegetation. How about using "49ka-alb" instead?

  Thank you for the suggestion, this has been changed to 49ka-alb throughout the manuscript now.

- According to which criteria did you implement the additional boundary conditions at a particular model year? It seems to be a bit subjective.

  We acknowledge that the choice of model year for implementing additional boundary conditions was subjective. As mentioned earlier, the stepwise approach was developed in parallel with the gradual availability of model capabilities. Our primary criterion was to ensure that there was no significant climate drift or instability in the trend between the application of one boundary condition and the next.

- Section 3.1: How about providing a little summary table for the global mean / hemispheric diagnostics for all experiments and reference to that instead of to Fig. 2 which does not explicitly show these numbers?

  We have now provided a table (Table A1) in the Supplementary Information.

| Experiments | PI | 49ka-full | 49ka-alb | 49ka-ic |
|---|---|---|---|---|
| Annual mean SAT (°C) | 15.41 | 12.71 (2.7) | 13.11 | 13.05 |
| DJF NH SAT (°C) | -4.75 | -11.15 (6.4) | -9.25 | -7.75 |
| JJA NH SAT (°C) | 14.95 | 12.35 (2.6) | 12.15 | 14.25 |
| DJF SH SAT (°C) | 2.85 | 1.15 (1.7) | 2.15 | 1.95 |
| JJA SH SAT (°C) | -4.85 | -7.25 (2.4) | -6.45 | -6.45 |
| Annual mean Ocean Temp (°C) | 4.32 | 3.3 (1.02) | 3.49 | 3.62 |
| Annual mean SST (°C) | 18.95 | 17.75 (1.2) | 17.65 | 17.75 |

**Table A1.** Global mean values for PI, 49ka-full, 49ka-alb, and 49ka-ic experiments. The values in the brackets show differences from PI.

- Fig. 2: Why is the significance testing only done for precipitation, and not for SAT/SST?

  It is done for all of them now. Please note the stippling now indicates non-significant changes.

[Figure]

Fig. R6. Annual mean anomalies of surface air temperature (SAT), sea surface temperature (SST), and precipitation for 49 ka climate (exp 49ka-full) compared to PI. Contours show seasonal mean 15\% sea ice concentration for (orange) 49 ka and (black) PI during DJF (dashed) and JJA (solid). Stippling indicates non-significant changes based on a Student's t-test at 95% confidence level. Stars in subpanels (a) and (b) indicate qualitative estimates of the changes between Greenland Interstadial (GI 13) and PI climate, as inferred from the proxy records (MDO1-2444,ODP-977A,MD95-2043,MD01-2443,MD95-2042 [Martrat et al., 2007]; ODP-1233 [Kaiser et al., 2005]; TR163-19 [Lea et al., 2000]; RC13-110 [Feldber and Mix, 2003]; ODP846B [Martinez et al., 2003]; TG-7 [Calvo et al., 2001]; MD97-2120 [Pahnke et al., 2003]; MD07-3128 [Caniupan et al., 2011]; EPICA Dronning Maud Land [EPICA community members]; WAIS [WAIS community members]; NGRIP [Huber et al., 2006, Kindler et al., 2014]; SO188-17286-1 [Lauterbach et al., 2020]. Dark (light) blue colors represent much colder (slightly colder) climate).

- Section 3.2: When discussing geopotential height (anomalies), I would also write this accordingly and not use the term "pressure" and/or insert "not shown" where required. Readers would look for corresponding (sea level) pressure figures which are not shown.

  Done.

- Section 3.2: For the jet stream, I would analyze upper tropospheric wind patterns such as uwind at 200 hPa. Wouldn't the Rossby wave response you mention be more evident e.g. at 500 hPa? In contrast to the statement on p. 8 l 174/175, I cannot see the planetary wave structure too clearly.

We agree that upper tropospheric wind patterns (e.g., 200 hPa) are typically used to investigate jet stream dynamics. However, in this study, our focus is primarily on surface-level changes to better link circulation shifts with the precipitation response.

That said, we have analysed upper-level winds (now included in the Supplementary Information). and geopotential height (GH) anomalies as well not shown) These show that the northeast–southeast tilted reorganisation of the jet stream is present at both surface and upper levels. However, we find that the precipitation anomalies correspond more closely with the surface-level GH patterns.

We have now clarified this in the manuscript, added the relevant upper-level figures to the SI (as Fig. A4), and expanded the discussion to highlight the consistency between surface and upper-level responses, while emphasising our focus on surface changes in this study.

[Figure]

Fig. R7: Zonal winds at 200 hPa (m/s) overlaid with wind vectors (m/s) in (left) DJF and (right) JJA for (top) PI and (middle) 49ka-full, and (bottom) 49ka-full minus PI.

- For the discussion in section 3.2 (p.7), Fig. A4 is quite clear and helpful. I would recommend to include it into the main part of the manuscript.

Thank you for the suggestion. It is now included as Fig. 5.

- Sections 3.2 to 3.4: I have some concerns regarding the section titles and the discussions in the respective

sections. I think that the wording has to be very precise to clearly distinguish between 1) the response to all 49 ka boundary conditions (b.c.) in the 49ka-full experiment and 2) the response in those experiments where only some b.c. have been prescribed. In that sense the section titles "Impact of the ice sheet topography" are partly incorrect unless you explicitly discuss the difference between 49ka-full and 49ka-ice, and even this comparison does not allow to attribute the changes you see exclusively to the ice sheets since you also modify salinity and the land-sea mask etc. The sentence p7 l. 153/154 is an example where to my opinion several aspects are mixed (land-sea distribution, ice-sheet height changes,...). The title of section 3.4 also needs some modification into e.g. „Impact of 49 ka boundary conditions on..."

The titles have been modified as follows.

3.2: Impact of 49ka boundary conditions on the circulation changes over North Atlantic and Eurasia.

3.3: Impact of 49ka boundary conditions on tropical and SH atmospheric circulation.

3.4: Impact of 49ka boundary conditions on Australian rainfall

- Section 3.2: I find it a bit hard to stay on track when following all the details between different seasons, different experiments, and different variables. Maybe starting the paragraph on p. 8 l. 158 with "In the DJF season" could create a clear structure and nicely contrast it with the paragraph starting in l. 169. This also holds for section 3.3 - try to make the structure immediately obvious for the reader (You do it nicely in section 3.4.).

  The structure for both sections was re-written to make it clearer.

- Section 3.2 and p.15 l. 271/272: moisture (flux convergence), specific humidity: Have you actually looked into these model results? Then it would be good to add "not shown" where appropriate. Otherwise it might be a bit speculative.

  Yes, we have checked this. "Not shown" is included where appropriate.

- p. 8 l. 164: I would also mention the strong cooling which seems to stand out in response to the albedo change.

  Done. Lines 196-198:

  In 49ka-alb, changing the vegetation and albedo has little impact compared to 49ka-co, except over northern North America (Figure 6d,f), where cooling is enhanced by 8° and drying increases by 25% due to the albedo change over the LIS.

- p. 8 l 174: "topography and height" - Isn't height included in topography?

  Rephrased.

- p. 8 l. 182 and 183: I am confused by the 20 Sv and cannot find this in the shaded part of Fig. 7a (dark reddish colors between about 15S and 10 N). Isn't the anomaly much larger according to the shading? (similar for the 11 Sv anomaly)

  We confirm that the anomalies are correct. That's the number we get for the maximum value averaged between 400 to 600hpa levels. We have now added the formula to the text.

$$\psi = \frac{2\pi R \cos\phi}{g} \int_0^p v\,dp \qquad (1)$$

where, R is the radius of the earth, $\phi$ is latitude, g is gravitational acceleration, v is the meridional wind and p is pressure.

- p. 8 l 184 Can you please include a sentence or reference for the link/mechanism between insolation and the Hadley cell strength?

  Here, we are referring to the shift in the ITCZ linked to peak summer insolation caused by higher obliquity. We have clarified this in the text.

- When comparing the Hadley cell among the different experiments, it would be helpful to have it in the main text and not in the supplement to allow direct comparison for the reader. The authors want to emphasize the role of the different boundary conditions, so it would be helpful to combine figures 7 and A6.

  Done.

[Figure]

Fig. R8. Atmospheric mass streamfunction (Sv) anomalies (shading) between (a,b) 49ka-full and PI, (c,d) 49ka-ic and PI, and (e,f) 49ka-ice and PI, for (left) DJF and (right) JJA. Contours (solid=positive; dashed=negative) are PI (black) absolute streamfunction values. The thick black and blue lines represent the zero contour for PI and 49ka experiments, respectively.

- p. 10 l. 190: "temperature gradient between the two hemispheres" - How about quantifiying this for all experiments and add it to the diagnostics table suggested above?

  Done. Added to Table A1 mentioned above.

- p. 10/11 l. 197/198: I suggest to rephrase this sentence. => "Temperature contrasts between hemispheres are amplified due to the introduction of ice sheet topography which induces localized warming in Siberia and strong Antarctic cooling, but also due to the replacement of ocean grid cells by land." It might also make things clearer if you explicitly specify the direction of the Hadley cell shift in relation to the hemisphere temperature contrast (shift towards the warmer hemisphere?)

  Done.

- p. 11: Since you already include the SH westerlies into Fig. 4, you could move the paragraph on SH westerlies to the end of section 3.2, discuss them there and keep a "tropical"-only focus in section 3.3.

  We thank the reviewer for this suggestion. We intended to maintain a meridional structure in the discussion, progressing from the Northern to the Southern Hemisphere. While we understand the value of keeping Section 3.3 focused solely on the tropics, we chose to include the Southern Hemisphere westerlies here to better link the Hadley cell, ITCZ, and SH westerly changes to the subsequent section on Australian rainfall, which is influenced by all three.

- section 3.4 p. 12 ll. 208/209: I do not fully agree with this statement. Doesn't Fig. 5c (49ka-co) already show the increase in Australian rainfall? You mention the strong obliquity change at 49 ka, so you might need to take this into account as well and not attribute everything to the ice sheets. I would start the section with l. 209 (slightly modified) and carefully phrase the following part in the light of the different boundary condtions, not just ice sheets, especially when referring to the 49ka-full figures (3,7) in the subsequent paragraphs. You could add for instance "in response to the 49 ka boundary conditions" to the sentence in p. 13 l. 223.

  Done.

- p. 14: AMOC strength during MIS3 seems to vary a lot among timeslices and models. A CCSM3 35 ka simulation initialized from LGM showed a weak AMOC (Merkel et al. doi:10.1016/j.quasci-rev.2009.11.006).

  We thank the reviewer for this suggestion. It has now been added as below.

  Lines 290-294:

  Model simulations also show considerable spread in AMOC strength across different MIS 3 timeslices and model setups. For instance, a Community Climate System Model version 3 (CCSM3) simulation at 35 ka,

initialized from the LGM, showed a ~40% reduction in AMOC strength (Merkel et la., 2010), while the same model produced a ~50% reduction at 44 ka, also initialised from the LGM state (Brandefelt et al., 2011). In contrast, the Norwegian Earth System Model (NorESM) simulations indicate a ~13% strengthening of the AMOC at 38 ka (Guo et al., 2019). The simulated AMOC in our model is strong at 49 ka (31 Sv) with a ~47\% increase relative to PI.

- p. 15 l. 275: What do you mean by "more dynamic"? The zonal response in Löfverström' study is also a dynamic one, isn't it?

  Yes. We have changed this now.

- p. 15 l. 279/280: There are studies which demonstrate the impact of glacial boundary conditions beyond the N. Atlantic sector (see doi indications above, but also for instance DiNezio et al. doi:10.1126/sci-adv.aat9658, Mohtadi et al. doi:10.1038/nature13196, Shi et al. 10.5194/cp-19-2157-2023).

  We have removed this part from the revised manuscript now.

- p. 15 l. 282: => "compared to our simulations"

  Modified.

- p. 15 vs. section 3: Some parts of the discussion on p. 15 have already been raised in section 3. Section 4 reads quite well, so you might consider to shorten section 3 and leave the interpretation for section 4.

  Thank you for this suggestion. We have applied it.

- section 4 Fig. 8: This is a nice summary which might already be worth to refer to during section 3. When averaging over the North Atlantic, is this all ocean grid points in a latitude range or did you chose some lat-lon box?

  It is a lat-lon box (80°W:0°E, 30°N:90°N). We have specified it now. We now also refer to Fig. 8 in section 3.

- section 4: I would recommend to modify the titles in bold of this section. I don't think these subsections can be clearly separated, and as noted above, phrasing should be done very carefully and with less focus on the ice sheets in the titles due to the experimental setup. It is appropriate to discuss the important role of the ice sheets, but the titles should be more general.

  Done. The new titles are:

  Changes in AMOC

  Changes in NH westerlies

  Changes in tropical atmospheric circulation and SH westerlies.

- p. 17 l.309: You might want to be a little bit more specific about the term "water availability" (soil moisture, atmospheric moisture content,...?).

  This study uses multiple hydroclimate records to assess the general wetness of Australia. Depending on the specific record, this may reflect soil moisture or atmospheric moisture. We would like to retain the term 'water availability' as it is.

- p. 17 l. 313: Wouldn't it make sense to also refer to your Fig. 3g to support your argument related to the insolation changes?

  Done.

- The Conclusions section is very short and does not mention MIS3 at all. It should be slightly rephrased in order to align better with the motivation and the main focus of the manuscript. Furthermore, the phrasing of the impacts of the ice sheets suggests that separate experiments have been conducted to isolate the respective impact of NH and SH ice sheets (LIS, AIS). Please make the wording more concise.

  It is now modified as below.

  Our results underscore the far-reaching effects of MIS 3 ice sheets on global atmospheric circulation and regional hydroclimate. We demonstrate that the presence of NH ice sheets during MIS 3 strengthened and shifted the North Atlantic westerlies equatorward by ~6° in DJF, leading to increased precipitation over the North Atlantic and Eurasian landmass in boreal summer. These ice sheet-induced changes extend

meridionally, with a southward shift of the NH Hadley cell and ITCZ during DJF, resulting in increased Australian rainfall. Although our experiments do not isolate the individual roles of the Laurentide and Antarctic ice sheets, simulations with MIS 3 topography suggest that the combined ice sheet configuration also contributes to an equatorward shift of the SH westerlies and SH Hadley cell. Overall, our findings underscore the importance of including realistic MIS 3 boundary conditions when assessing ice sheet–atmosphere interactions on atmospheric circulation.

- Data availability: Will also the code, e.g. to calculate the atmospheric mass streamfunction, be made available? Which density has been used for the calculation in Sv?

  Yes, the code would be made available. In addition, we provide the details of our method for computing the atmospheric mass stream function ($\psi$), which follows standard conventions. Specifically:

$$\psi = \frac{2\pi R \cos\phi}{g} \int_0^p v \, dp \tag{1}$$

where, R is the radius of the earth, $\phi$ is latitude, g is gravitational acceleration, v is the meridional wind and p is pressure.

- To my knowledge, in a student's t-test, you would call 95% the confidence level and 5% the significance level. Please correct the corresponding text and figure captions.
  Done.

- Fig. A1 caption mentions „simulated ice sheet", but according to the Methods section, ice sheets are pre-scribed as constant forcing in this study.

  Yes. Corrected.

- Fig. A3 mentions „North Atlantic gyre strength". Is it the barotropic streamfunction?

  Yes, it is the barotropic stream function. We have now mentioned this in the text now.

The below technical corrections have now been either directly implemented, or a response is given.

**Technical corrections**

- Please double-check all acronyms, not all of them have been explained or explained where they appear for the first time (e.g. CMIP6 p. 3 l. 63, CASA-CNP p. 3 l. 67)

- p. 1 l. 3 "simulate a glacial climate" => "simulate the glacial climate"

- p.1 l. 14 "by their extent" => "by the ice sheet extent" (and also height?)

- p. 1 l.19 "on the atmospheric"

- p. 2 l. 45 Menviel et al., 2020 => Menviel et al., 2020 and references therein

- Section 2.1 l. 67: "uses" => "consists in"

- p. 3 l. 81/82: => "with only the orbital parameters… and GHG concentrations … being set to 49 ka va- lues…"

- p. 4: Tab. 1: I would just put 49 ka instead of 49ka-full.

- p. 6 l. 1 => "divides the tropical precipitation amount"

- p. 6 l. 4 "within the same range" => „over the same vertical/pressure range". Is there a reference for cho- sing 400-600 hPa?

- Tab. 2: caption: "in all" => "in the different 49 ka experiments"; "in the full time-series shown in" => "as shown in the timeseries of Fig. A1"

- Fig. 1 caption: a), b) missing; "per grid cell" => "of each grid cell"

- Fig. 2: Sea-ice edges are very hard to see in Fig. 2b. How about chosing polar stereographic projection instead?

  We thank the reviewer for the suggestion. To maintain consistency across all subpanels in Fig. 2, we prefer to retain the current projection. However, we have adjusted the figure to improve visibility by making the sea-ice edges bolder and more distinguishable.

- Fig. 2 caption: "ocean ice" => "sea ice". "Contours are" => "Contours show". I would also start the caption with "Annual mean anomalies of…"

- l. 125: => "by 5 to 6° latitude"

- l. 127: NADW => Do you miss "formation" here?

- p. 7 l. 137: I guess you are referring to the meridional temperature gradient here. Please add.

- p. 7 l. 138: => "shift by ~6°"

- Fig. 3 caption: I would start with the caption with "Anomalies between 49ka-full and PI for (a,b) SAT (°C)…"

- Fig. 3c-f: What happens at 0°E in the vector plots? Have you also looked at higher levels to avoid the plotting conflict with topography?

  Fig.3c-f is fixed now. We have also now included the upper-level winds (Fig. R6 above) in SI as Fig. A5.

- p. 14 l. 269: => "landmasses"

- Fig. 8: Since you have a), b) in the Figure, you could adjust the caption accordingly. You could also make the

lat-lon specifications in brackets consistent (hyphen, colon) and make the Figure and caption consistent (Europe/Eurasia).

Done.

- Fig. A1: The caption does not mention the lower panels and does not have references to "left panels", "middle panels" etc. You could also add a), b), labels. Please also mention what is shown, I guess it should read "Timeseries of anomalies of annual mean surface air temperature...".

We apologies for missing these details and thank the reviewer for noting it. The caption has now been modified as in Fig. R5 above.

- Fig. A2: What exactly is shown here? Have you taken the SW downward radiation at the top of the atmosphere/model?

Yes, it is the incoming shortwave radiation at the top of the atmosphere.

- Fig. A3: Please mention in the caption that the white areas mark grid cells with continental ice, and that for the 49 ka simulation, continental outlines are shown based on the adjusted land-sea mask. I would also write "...the 49ka_full to PI difference".

Done.

- Fig. A5: caption incomplete: It is missing vector descriptions.

Done. Apologies for missing this.